# The Ultimate Combo: Boosting Adversarial Example Transferability by Composing Data Augmentations

## Abstract

Transferring adversarial examples (AEs) from surrogate machine-learning (ML) models to evade target models is a common method for evaluating adversarial robustness in black-box settings. Researchers have invested substantial efforts to enhance transferability. Chiefly, attacks leveraging data augmentation have been found to help AEs generalize better from surrogates to targets. Still, prior work has explored a limited set of augmentation techniques and their composition. To fill the gap, we conducted a systematic study of how data augmentation affects transferability. Particularly, we explored ten augmentation techniques of six categories originally proposed to help ML models generalize to unseen benign samples, and assessed how they influence transferability, both when applied individually and when composed. Our extensive experiments with the ImageNet and CIFAR-10 dataset showed that simple color-space augmentations (e.g., color to greyscale) outperform the state of the art when combined with standard augmentations, such as translation and scaling. Additionally, except for two methods that may harm transferability, we found that composing augmentation methods impacts transferability monotonically (i.e., more methods composed $\rightarrow$ $\geq$transferability)—the best composition we found significantly outperformed the state of the art (e.g., 95.6% vs. 92.0% average transferability on ImageNet from normally trained surrogates to other normally trained models). We provide intuitive, empirically supported explanations for why certain augmentations fail to improve transferability.

## 1 Introduction

Adversarial examples (AEs)—variants of benign inputs minimally perturbed to induce misclassification at test time—have emerged as a profound challenge to machine learning (ML) (Biggio et al., 2013; Szegedy et al., 2014), calling its use in security- and safety-critical systems into question (e.g., Eykholt et al. (2018)). Many attacks have been proposed to generate AEs in white-box settings, where adversaries are familiar with all the particularities of the attacked model (Papernot et al., 2016). By contrast, black-box attacks enable evaluating the vulnerability of ML in realistic settings, without access to the model (Papernot et al., 2016).

Attacks exploiting the transferability-property of AEs (Szegedy et al., 2014) have received special attention. Namely, as AEs produced against one model are often misclassified by others, transferability-based attacks produce AEs against surrogate (a.k.a. substitute) white-box models to mislead black-box ones. To measure the risk of AEs in black-box settings accurately, researchers have proposed varied methods to enhance transferability (e.g., Lin et al. (2020); Liu et al. (2017)).

Notably, attacks using data augmentation, such as translations (Dong et al., 2019) and scaling of pixel values (Lin et al., 2020), as a means to improve the generalizability of AEs across models have accomplished state-of-the-art transferability rates. Still, previous transferability-based attacks have studied only four augmentation methods (see Section 3.1), out of many proposed in the data-augmentation literature (Shorten & Khoshgoftaar, 2019), primarily for reducing model overfitting. Hence, the extent to which different data-augmentation types boost transferability, either individually or when combined, remains largely unknown.

To fill the gap, we conducted a systematic study of how augmentation methods influence transferability. Specifically, alongside techniques considered in previous work, we studied how ten augmentation techniques pertaining to six categories impact transferability when applied individually or composed (Section 3). Integrating augmentation methods into attacks via a flexible framework we propose (Algorithm 1), we conducted extensive experiments using an ImageNet-compatible dataset, CIFAR-10 (Krizhevsky, 2009), and 16 models, and measured transferability in diverse settings, including with and without defenses (Sections 4 and 5). Our results offer several interesting insights:

- Simple color-space augmentations outperform state-of-the-art transferability-based attacks when composed with standard augmentations (Section 5.1).

- Transferability has a mostly monotonic relationship with data-augmentation techniques. Except for two augmentation methods that may harm transferability, composing additional augmentation methods either improves of preserves transferability (Section 5.2).

- Out of $2^7$ compositions explored, the best composition we found, UltimateCombo, outperforms state-of-the-art attacks by a large margin (Section 5.3).

- We show empirical support to conjectures we raise concerning when data-augmentation techniques may be counterproductive to transferability (Section 5.4).

## 2 Background and Related Work

***Evasion Attacks*** Many evasion attacks assume adversaries have white-box access to models—i.e., adversaries know models' architectures and weights (e.g., Goodfellow et al. (2015); Szegedy et al. (2014); Carlini & Wagner (2017)). These typically leverage first- or second-order optimizations to generate AEs models would misclassify. For example, given an input $x$ of class $y$, model weights $\theta$, and a loss function $J$, the Fast Gradient Sign method (FGSM) of Goodfellow et al. (2015), crafts an AE $\hat{x}$ using the loss gradients $\nabla_x J(x, y, \theta)$:

$$\hat{x} = x + \epsilon * \text{sign}(\nabla_x J(x, y, \theta))$$

where $\text{sign}(\cdot)$ maps real numbers to -1, 0, or 1, depending on their sign. Following FGSM, researchers proposed numerous advanced attacks. Notably, iterative FGSM (I-FGSM) of Kurakin et al. (2017b) performs multiple gradient-ascent steps, updating $\hat{x}$ iteratively to evade models:

$$\hat{x}_{t+1} = \text{Proj}_x^\epsilon \left( \hat{x}_t + \alpha \cdot \text{sign} \left( \nabla_x J \left( \hat{x}_t, y, \theta \right) \right) \right)$$

where $\text{Proj}_x^\epsilon(\cdot)$ projects the perturbation into $\ell_\infty$-norm $\epsilon$-ball centered at $x$, $\alpha$ is the step size, and $\hat{x}_0 = x$. The attacks we study in this work are based on I-FGSM.

In practice, adversaries often lack white-box access to victim models. Hence, researchers studied black-box attacks in which adversaries may only query models. Certain attack types, such as score- and boundary-based attacks perform multiple queries, often around several thousands, to produce AEs (e.g., Brendel et al. (2018); Ilyas et al. (2019)). By contrast, attacks leveraging *transferability* (e.g., Goodfellow et al. (2015); Szegedy et al. (2014)) avoid querying victim models, and use surrogate white-box models to create AEs that are likely misclassified by other black-box ones.

Attempts to explain the transferability phenomenon attribute it to gradient norm of the target model (i.e., its susceptibility to attacks), the smoothness of classification boundaries, and, primarily, the alignment of gradient directions between the surrogate and target models (Demontis et al., 2019; Yang et al., 2021). Said differently, for AEs to transfer, the gradient directions of surrogates need to be similar to those of target models (i.e., attain high cosine similarity).

Enhancing transferability is an active research area. Some methods integrate momentum into attacks such as I-FGSM to avoid surrogate-specific optima and saddle points that may hinder transferability (e.g., Dong et al. (2018); Wang & He (2021)). Others employ specialized losses, such as reducing the variance of intermediate activations (Huang et al., 2019) or the mean loss of model ensembles (Liu et al., 2017), to enhance transferability. Lastly, a prominent family of attacks leverages data augmentation to enhance AEs' generalizability between models. For instance, Dong et al. (2019) boosted transferability by integrating random translations into I-FGSM. Evasion attacks incorporating data augmentation attain state-of-the-art transferability rates (Lin et al., 2020; Wang et al.,

---

**Algorithm 1** MI-FGSM with data augmentation

---

**Input:** Benign sample $x$; ground-truth label $y$; loss function $J(\cdot)$; model parameters $\theta$; iterations #
$T$; momentum parameter $\mu$; perturbation size $\epsilon$; data-augmentation method $D(\cdot)$.
**Output:** Adversarial example $\hat{x}$

  1: $\alpha = \epsilon/T$
  2: $\hat{x}_0 = x$                                  # Initialize adversarial example
  3: $g_0 = 0$                                    # Initialize momentum
  4: **for** $t = 0$ to $T - 1$ **do**
  5:     $\bar{g}_{t+1} = \frac{1}{m} \sum_{i=0}^{m-1} \nabla_x \left( J \left( D(\hat{x}_t)_i, y, \theta \right) \right)$   # Expected loss gradient on augmented samples
  6:     $g_{t+1} = \mu \cdot g_t + \frac{\bar{g}_{t+1}}{\|\bar{g}_{t+1}\|_1}$   # Gradient with momentum
  7:     $\hat{x}_{t+1} = \text{Proj}_x^\epsilon \left( \hat{x}_t + \alpha \cdot \text{sign} \left( g_{t+1} \right) \right)$   # Update adversarial example
  8: **return** $\hat{x} = \hat{x}_T$

---

2021a). Nonetheless, prior work has only considered a restricted set of augmentation methods for boosting transferability. By contrast, we aim to investigate the role of data augmentation at enhancing transferability more systematically, by exploring how a more comprehensive set of augmentation types and their compositions affect transferability.

***Defenses*** Various defenses have been proposed to mitigate evasion attacks. Adversarial training—a procedure integrating correctly labeled AEs in training—is one of the most practical and effective methods for enhancing adversarial robustness (e.g., Goodfellow et al. (2015); Tramèr et al. (2018)). Other defense methods sanitize inputs prior to classification (e.g., Guo et al. (2018)); attempt to detect attacks (see Tramer (2022)); or seek to certify robustness in $\epsilon$-balls around inputs (e.g., Cohen et al. (2019); Salman et al. (2019)). Following standard practices in the literature (Wang et al., 2021a), we evaluate transferability-based attacks against a representative set of these defense.

## 3 DATA AUGMENTATION FOR ENHANCING TRANSFERABILITY

Data augmentation is traditionally used in training, to reduce overfitting and improve generalizability (Shorten & Khoshgoftaar, 2019). Inspired by this use, transferability-based attacks adopted data augmentation to limit overfitting to surrogate models and produce AEs likely to generalize and be misclassified by victim models. Algorithm 1 depicts a general framework for integrating data augmentation into I-FGSM with momentum (MI-FGSM). In the framework, a method $D(\cdot)$ augments the attack with $m$ variants of the estimated AE at each iteration. Consequently, the adversarial perturbation found by the attack increases the expected loss over transformed counterparts of the benign sample $x$ (i.e., the distribution set by $D(\cdot)$ given $x$). Note that $D(\cdot)$'s output may include $x$.

The framework in Algorithm 1 is flexible, and can admit any data-augmentation method. We use it to describe previous attacks employing data augmentation and to systematically explore new ones. Next, we detail previous attacks, describe data augmentation methods we adopt for the first time to enhance transferability, and explain how these can be combined for best performance.

### 3.1 PREVIOUS ATTACKS LEVERAGING DATA AUGMENTATION

Previous work explored the following augmentation methods to set $D(\cdot)$.

***Translations*** Using random translations of inputs, Dong et al. (2019) proposed a translation-invariant attack to promote transferability. They also offered an optimization to reduce the attack's time and space complexity by simply convolving the model's gradients (w.r.t. non-translated inputs) with a Gaussian kernel. While we use this optimization in the implementation for the interest of efficiency, we highlight that the attack can be well-captured by our framework.

***Diverse Inputs*** Xie et al. (2019) proposed a size-invariant attack. Their augmentation procedure samples random crops from $\hat{x}_t$ that are later resized per the model's input dimensionality.

***Scaling Pixels*** Lin et al. (2020) showed that adversarial perturbations invariant to scaling pixel values transfer with higher success between deep neural networks (DNNs). In their case, $D(\cdot)$ produces $m$ samples such that $D(x)_i = \frac{x}{2^i}$ for $i \in \{0, 1, ..., m-1\}$, where $m$=5 by default.

***Admix*** Wang et al. (2021a) assumed that the adversary has a gallery of images from different classes and adopted augmentations similar to *MixUp* (Zhang et al., 2018a). For each sample $x'$ from the gallery, Admix augments attacks with $m$ (typically set to 5) samples, such that $D(x, x')_i = \frac{1}{2^i} \cdot (\hat{x}_t + \eta \cdot x')$, where $i \in \{0, 1, ..., m-1\}$, and $\eta \in [0, 1]$ is set to 0.2 by default. Notably, Admix degenerates to pixel scaling when $\eta = 0$.

The leading transferability-based attacks compose *(1)* diverse inputs, scaling, and translations (Lin et al.'s (2020) DST-MI-FGSM attack, and Wang & He's (2021) DST-VMI-FGSM attack that also tunes the gradients' variance); or *(2)* Admix, diverse inputs, and translation (Wang et al.'s (2021a) Admix-DT-MI-FGSM attack). We describe how the compositions operate in Section 3.3.

## 3.2 NEW AUGMENTATIONS FOR ENHANCING TRANSFERABILITY

While prior work studied the effect of spatial transformations (i.e., translations and diverse inputs), pixel scaling, and mixing on transferability, a substantially wider range of data-augmentation methods exist. Yet, the impact of these on transferability remains unknown. To fill the gap, we examined Shorten & Khoshgoftaar's (2019) survey on data augmentation for reducing overfitting in deep learning and identified ten representative methods of six categories that may boost transferability. We present them in what follows, one category at a time.

***Color-space Transformations*** Potentially the simplest of all augmentation types are those applied in color-space. Given images represented as three-channel tensors, methods in this category manipulate pixel values only based on information encoded in the tensors. We evaluate four color-space transformations. First, we consider *color jitter (CJ)*, which applies random color manipulation (Wu et al., 2015). Specifically, we consider random adjustments of pixel values within a pre-defined range in terms of hue, contrast, saturation, and brightness around original values. Second, we evaluate *fancy principle component analysis (fPCA)*. Used in AlexNet (Krizhevsky et al., 2017), fPCA adds noise to the image proportionally to the variance in each channel. Given an RGB image, fPCA adds the following quantity to each image pixel:

$$[\mathbf{p}_1, \mathbf{p}_2, \mathbf{p}_3] [\alpha_1 \lambda_1, \alpha_2 \lambda_2, \alpha_3 \lambda_3]^T ,$$

where $p_i$ and $\lambda_i$ are the $i^{\text{th}}$ eigenvector and eigenvalue of the of $3 \times 3$ covariance matrix of RGB pixels, respectively, and $\alpha_i$ is sampled once per image from Gaussian distribution $\mathcal{N}(0, 0.1)$. Third, we test *channel shuffle (CS)*. Included in ShuffleNet training (Zhang et al., 2018b), CS simply swaps the orders of the image's RGB channels at random. Last, but not least, we consider *greyscale (GS)* augmentations. This simple augmentation converts images into greyscale (replicating it three times to obtain an RGB representation). Mathematically, the conversion is calculated by $\omega_R \cdot x^R + \omega_G \cdot x^G + \omega_B \cdot x^B$, where $x^R$, $x^G$, and $x^B$, correspond to the RGB channels, respectively, and $\omega_R, \omega_G,$ and $\omega_B$, all $\in [0, 1]$, denote the channel weights, and sum up to 1.

***Random Erasing*** Inspired by dropout regularization, *random erasing (RE)* helps ML models focus on descriptive features of images and promote robustness to occlusions (Zhong et al., 2020). To do so, randomly selected rectangular regions in images are replaced by masks composed of random pixel values. Similarly to RE, *CutOut* masks out regions of inputs to improve DNNs' accuracy (De-Vries & Taylor, 2017). The main difference from e is that CutOut uses fixed masking values, and may perform less aggressive masking when selected regions lie outside the image.

***Kernel Filters*** Convolving images with kernels of different types can produce certain effects, such as blurring (via Gaussian kernels), sharpening (via edge filters), or edge enhancement. We study the effect of *sharpening (Sharp)* on transferability with edge-enhancement filters.

***Mixing Images*** As a form of vicinal risk minimization, some augmentation methods mix images together, creating virtual examples for training. MixUp, the cornerstone behind Admix, computes weighted sums of images (Zhang et al., 2018a). By contrast, we consider *CutMix*, which replaces a region within one image with a region from another image picked from a gallery (Yun et al., 2019).

***Neural Transfer*** Augmentations using *neural transfer (NeuTrans)* preserve image semantics while changing their style. We use Gatys et al.'s (2015) generative model to transfer image styles to that of Picasso's 1907 self-portrait.

***Meta-learning-inspired Augmentations*** Meta-learning is a subfield of ML studying how ML algorithms can optimize other learning algorithms (Hospedales et al., 2021). In the context of data augmentation, algorithms such as *AutoAugment* have been proposed to train controllers to select an appropriate augmentation method to avoid overfitting (Cubuk et al., 2019). We use the pre-trained AutoAugment controller, encoded as a recurrent neural network, to select augmentation methods and their magnitude from a set of 13 augmentation methods.

## 3.3 COMPOSING AUGMENTATIONS

There are two ways to compose data-augmentation methods in attacks, namely: *parallel* and *serial* composition. Figure 1 in Appendix A illustrates both. In parallel composition, each augmentation method is applied independently on the input, and their outputs are aggregated by taking their union to augment attacks (i.e., as $D(\cdot)$'s output). By contrast, serial composition applies augmentation methods sequentially, one after the other, where the first method operates on the original sample, and each subsequent augmentation function operates on its predecessor's outputs. Consequently, serial composition leads to an *exponential* growth in the number of samples, while parallel composition leads to a *linear* growth. DST-MI-FGSM and Admix-DT-MI-FGSM use serial composition. By contrast, we consider a substantially larger number of augmentation methods, which may lead to prohibitive memory and time requirements in the case of serial composition. Additionally, because the order of applying certain augmentations matters (e.g., GS then CutMix leads to different outcome that CutMix followed by GS), exploring a meaningful number of serial compositions (out of an order of 10! possibilities) becomes virtually impossible. Accordingly, we mainly consider parallel composition between data-augmentation methods. We *only* serially compose translations, scaling, and diverse inputs, for consistency with prior work (e.g., Wang et al. (2021a)). We tested a few serial compositions between new augmentation methods we consider and found they were significantly outperformed by their parallel counterparts. While non-exhaustive, this hints that serially composing augmentations may not be a promising direction for enhancing transferability.

## 4 EXPERIMENTAL SETUP

Now we turn to the setup of our experiments, including the data, models, and attack configurations.

***Data*** We used an ImageNet-compatible dataset[1] and CIFAR-10 for evaluation, per common practice (e.g., (Dong et al., 2019; Yang et al., 2021)). The former contains 1,000 images, originally collected for the NeurIPS 2017 adversarial ML competition. For the latter, we sampled 1,000 images, roughly balanced between classes, from the test set.

***Models*** We used 16 DNNs to transfer attacks from (as surrogates) and to (as targets)—six for CIFAR-10 and ten for ImageNet. All CIFAR-10 models and six of the ImageNet models were normally trained, while the other four ImageNet models were adversarially trained. To facilitate comparison with prior work, we included models that are widely used for assessing transferability (e.g., (Wang et al., 2021a; Yang et al., 2021)). Furthermore, to ensure that our findings are general, we included models covering varied architectures, including Inception, ResNet, VGG, DenseNet, and MobileNet. Appendix B provides more details about the models.

***Attack Parameters*** We tested standard attack configurations, in line with prior work (Wang et al., 2021a; Yang et al., 2021). Namely, we evaluated untargeted MI-FGSM-based attacks, bounded in $\ell_\infty$-norm. We validated findings with varied perturbation norms. For ImageNet, unless stated otherwise, we tested $\epsilon = \frac{16}{255}$, but also experimented with $\epsilon \in \{\frac{8}{255}, \frac{24}{255}\}$. For CIFAR-10, we experimented with $\epsilon \in \{0.02, 0.04\}$. We quantified attack success via *transferability rates*—the percentages of attempts at which AEs created against surrogates were misclassified by victims. As baselines, we used three state-of-the-art transferability-based attacks: DST-MI-FGSM, DST-VMI-FGSM, and Admix-DT-MI-FGSM (see Section 3.1). Appendix C reports the parameters used in

---

[1]https://bit.ly/3fq4pN6

attacks and augmentation methods. Appendix D discusses attacks we considered but excluded from experiments.

# 5 EXPERIMENTAL RESULTS

This section summarizes our findings. We start by evaluating individual augmentation methods and standard combinations with scaling, diverse inputs, and translations (Section 5.1). We then turn to analyzing *all possible* compositions between different augmentation types to assess whether transferability typically improves when considering additional augmentations (Section 5.2). Our analysis helped us identify the best performing composition for boosting transferability, denoted by ULTIMATECOMBO, outperforming state-of-the-art attacks. Section 5.3 reports rigorous comparisons between ULTIMATECOMBO and the baselines, including against defended models. Finally, we help develop intuition for when augmentations may or may not help improve transferability (Section 5.4).

## 5.1 COLOR-SPACE AUGMENTATIONS SIGNIFICANTLY ADVANCE THE STATE OF THE ART

Initially, we evaluated transferability integrating a single augmentation at a time in attacks, or when composing individual augmentations with diverse inputs, scaling, and translation (DST), as is standard (Lin et al., 2020; Wang et al., 2021a). We found that considering each of the ten augmentations individually does *not* lead to competitive performance with the baselines (Table 9 in Appendix E). However, composing individual augmentations with DST enhanced transferability markedly (Table 10 in Appendix E). Surprisingly, augmentations in color-space fared particularly well, outperforming the baselines and advanced augmentation methods (e.g., AutoAugment) in most cases.

Composing GS with DST (GS-DST-MI-FGSM attack) performed best in this setting. Table 1 reports the transferability rates from four normally trained models to other models on ImageNet (see Table 11 in Appendix F for more details). It can be immediately seen that GS-DST-MI-FGSM attains higher transferability than the baselines (93.6% vs. ≤92.0%, on avg.). This held also when considering different perturbation norms on ImageNet, where GS-DST-MI-FGSM outperformed the baselines with sometimes higher margin (e.g., 75.9% vs. ≤70.8% on avg. with $\epsilon = \frac{8}{255}$; see Table 13 in Appendix F). GS-DST-MI-FGSM also outperformed the baselines on CIFAR-10, when transferring AEs to normally trained DNNs of different architectures, with perturbation norms $\epsilon$=0.02 (74.9% vs. ≤71.5% avg. transferability rate) and $\epsilon$=0.04 (92.1% vs. ≤89.6% avg. transferability rate). Tables 14 and 15 in Appendix G show the detailed CIFAR-10 results.

| Model | Attack | Inc-v3 | Inc-v4 | Res-50 | Res-101 | Res-152 | IncRes-v2 |
|---|---|---|---|---|---|---|---|
| | MAXBASELINE | **100.0** | 94.7 | 90.7 | 88.9 | 89.1 | 92.6 |
| Inc-v3 | GS-DST-MI-FGSM | *100.0* | *95.6* | *93.7* | *91.8* | *90.9* | *94.9* |
| | ULTIMATECOMBO | **100.0** | **98.0** | **95.1** | **94.3** | **92.7** | **97.1** |
| | MAXBASELINE | 95.3 | **100.0** | 91.0 | 89.9 | 88.4 | 93.5 |
| Inc-v4 | GS-DST-MI-FGSM | *96.5* | *100.0* | *94.1* | *92.5* | *93.0* | *95.4* |
| | ULTIMATECOMBO | **98.1** | *99.9* | **94.8** | **95.0** | **94.6** | **96.8** |
| | MAXBASELINE | 88.3 | 85.0 | 97.6 | **99.9** | 96.9 | 87.2 |
| Res-101 | GS-DST-MI-FGSM | *89.0* | *84.8* | *97.6* | *99.8* | *97.7* | *87.6* |
| | ULTIMATECOMBO | **93.0** | **90.4** | **98.1** | 99.7 | **97.8** | **91.8** |
| | MAXBASELINE | 95.8 | 94.7 | 94.0 | 92.9 | 92.9 | 99.8 |
| IncRes-v2 | GS-DST-MI-FGSM | *96.5* | *95.6* | *95.5* | *94.2* | *94.7* | **100.0** |
| | ULTIMATECOMBO | **98.2** | **97.1** | **96.3** | **96.5** | **95.7** | **100.0** |

Table 1: Transferability rates (%) on ImageNet, from normally trained surrogates (rows) to normally trained targets (columns). All attacks are black-box, except for when the surrogate and target models are the same. MAXBASELINE is the best performing of the three baselines.

The same trends held when transferring AEs to adversarially trained models. Here, we transferred AEs from individual, normally trained DNNs (Table 2 reports a summary, and Table 12 shows complete results), as well as an ensemble of DNNs (Table 3) used to boost transferability further (Liu et al., 2017), finding that GS-DST-MI-FGSM attained better transferability than the baselines. Overall, according to a paired t-test, the differences between GS-DST-MI-FGSM and the baselines across different surrogate and target models were statistically significant ($p <$0.01).

| Model | Attack | Inc-v3$_{adv}$ | Inc-v3$_{ens3}$ | Inc-v3$_{ens4}$ | IncRes-v2$_{ens}$ |
|---|---|---|---|---|---|
| Inc-v3 | MAXBASELINE | 84.6 | 84.8 | 83.5 | 70.8 |
| | GS-DST-MI-FGSM | *87.3* | *88.5* | *85.5* | *72.2* |
| | ULTIMATECOMBO | **88.2** | **88.7** | **86.7** | **72.6** |
| Inc-v4 | MAXBASELINE | 84.3 | 86.0 | 83.0 | 74.6 |
| | GS-DST-MI-FGSM | *87.6* | **89.5** | *87.2* | *78.0* |
| | ULTIMATECOMBO | **88.6** | 89.4 | **88.4** | **78.2** |
| Res-101 | MAXBASELINE | 82.0 | 83.0 | 80.9 | 72.5 |
| | GS-DST-MI-FGSM | *82.3* | *83.7* | *81.1* | *73.2* |
| | ULTIMATECOMBO | **83.5** | **86.7** | **82.8** | **76.8** |
| IncRes-v2 | MAXBASELINE | 89.0 | 89.0 | 88.7 | 87.1 |
| | GS-DST-MI-FGSM | *91.1* | *92.2* | *90.0* | *88.2* |
| | ULTIMATECOMBO | **92.2** | **92.6** | **92.0** | **88.5** |

Table 2: Transferability rates (%) on ImageNet, from normally trained surrogates (rows) to adversarially trained targets (columns). MAXBASELINE is the best performing of the three baselines.

| Attack | Inc-v3$_{adv}$ | Inc-v3$_{ens3}$ | Inc-v3$_{ens4}$ | IncRes-v2$_{ens}$ |
|---|---|---|---|---|
| DST-MI-FGSM | 89.0 | 90.0 | 87.6 | 82.4 |
| Admix-DT-MI-FGSM | 90.1 | 90.5 | 89.4 | 84.7 |
| GS-DST-MI-FGSM | *92.4* | *93.5* | *92.5* | *88.7* |
| ULTIMATECOMBO | **93.6** | **95.2** | **93.7** | **91.2** |

Table 3: Transferability rates (%) on ImageNet, from an ensemble of normally trained surrogates (containing Inc-v4, Res-50, Res-101 and Res-152) to adversarially trained target models. DST-VMI-FGSM was excluded due to resource constrains.

Finally, we evaluated attack run-times, finding that, despite investing no effort to improve its efficiency, GS-DST-MI-FGSM is at least ×1.14 more time-efficient than Admix-DT-MI-FGSM and DST-VMI-FGSM, on avg. (Table 16 in Appendix I). Still, we denote that, since transferability-based attacks generate AEs offline, and only once per surrogate model, as long as an attack is not prohibitively slow, attack run-time is a marginal consideration for selecting an attack compared to transferability rates.

## 5.2 THE MONOTONICITY OF TRANSFERABILITY WHEN ADDING AUGMENTATIONS

We wanted to evaluate whether transferability is monotonic in the number of augmentation types considered—i.e., whether composing more techniques increases, or at least does not harm, transferability. To this end, we selected the best performing augmentation method of each of the six categories presented in Section 3.2 as well as DST-MI-FGSM, and evaluated all $2^7$ (=128) compositions possible (per Section 3.3). More precisely, we tested every possible combination of GS, CutOut, Sharp, NeuTrans, AutoAugment, Admix, and DST-MI-FGSM. Given a composition, we produced AEs against the Inc-v3 ImageNet DNN as surrogate, and computed the expected transferability rate against all other nine ImageNet DNNs, both normally and adversarially trained. Then, for every pair of attacks differing only in whether a single augmentation method was incorporated in the composition, we tested whether adding the augmentation method improved transferability.

The results reflected a mostly monotonic relationship between transferability and augmentations. Except for NeuTrans and Sharp, which sometimes harmed transferability when considered within a composition, adding augmentation method increased or preserved transferability. Figure 2 in Appendix H summarizes the results. Notably, comparing all compositions enabled us to find that a composition of all seven augmentation methods except for NeuTrans attained the best transferability. We call this composition the ULTIMATECOMBO.

## 5.3 THE MOST EFFECTIVE COMBINATION

We evaluated ULTIMATECOMBO extensively, testing transferability to normally and adversarially trained DNNs. As shown in Table 1, ULTIMATECOMBO obtained higher transferability to normally

trained models than the baselines (95.6% vs. $\leq$92.0% avg. transferability) and GS-DST-MI-FGSM, when normally trained models were used as surrogates. This holds across different values of $\epsilon$ (Table 13), and on the CIFAR-10 dataset with different architectures (Tables 14 and 15).

Furthermore, ULTIMATECOMBO achieved the best performance also when transferring attacks from normally trained to adversarially trained DNNs (Table 2; 86.0% vs. $\leq$82.7% avg. transferability). Transferring AEs crafted by ULTIMATECOMBO using an ensemble of models increased transferability further (Table 3; 93.4% avg. transferability). Per a paired t-test, the differences between ULTIMATECOMBO and the baselines over all pairs of surrogates and targets considered are statistically significant ($p <$0.01).

Besides adversarially trained models, we evaluated ULTIMATECOMBO's transferability against five defenses. Two defenses, bit reduction (Bit-Red) (Xu et al., 2018) and neural representation purification (NRP) (Naseer et al., 2020), transform inputs to sanitize adversarial perturbations. Two others, randomized smoothing (RS) (Cohen et al., 2019) and randomized smoothing with adversarial training (ARS) (Salman et al., 2019) offer provable robustness guarantees. Finally, TRS leverages an ensemble of smooth DNNs trained to have misaligned gradients, to defend attacks (Yang et al., 2021). We evaluated all defenses except for TRS on ImageNet. We used the defenses with default parameters (see Appendix J), and transferred AEs crafted against an ensemble of normally trained models. Results are shown in Table 4. Similar to other settings, here too, ULTIMATECOMBO outperformed the baselines (66.8% vs. $\leq$63.9% avg. transferability). Following Yang et al. (2021), we tested TRS on CIFAR-10 with adversarial perturbation norms $\epsilon \in \{0.02, 0.04\}$. ULTIMATECOMBO did best against this defense as well (Table 5).

| Attack | Bit-Red | NRP | RS | ARS |
|---|---|---|---|---|
| DST-MI-FGSM | 85.3 | *40.7* | 84.2 | 39.8 |
| Admix-DT-MI-FGSM | *86.4* | 39.4 | *86.6* | *43.0* |
| ULTIMATECOMBO | **87.5** | **47.7** | **88.4** | **43.5** |

Table 4: Transferability rates (%) from an ensemble of normally trained surrogates (Inc-v4, Res-50, Res-101 and Res-152) to models defended by provable methods or input transformations. DST-VMI-FGSM was excluded due to resource constrains.

| Epsilon | Admix-DT-MI-FGSM | DST-MI-FGSM | DST-VMI-FGSM | GS-DST-MI-FGSM | ULTIMATECOMBO |
|---|---|---|---|---|---|
| 0.02 | 21.5 | 23.1 | 18.9 | *25.1* | **27.4** |
| 0.04 | 36.2 | 41.3 | 36.3 | *47.8* | **49.4** |

Table 5: Transferability rates (%) on CIFAR-10 from a normally trained VGG surrogate DNN to an ensemble of Res DNNs trained via TRS.

Lastly, due to composing more augmentations, ULTIMATECOMBO is slower than DST-MI-FGSM, GS-DST-MI-FGSM, and Admix-DT-MI-FGSM. However, it is $\times$2.44 faster than DST-VMI-FGSM at producing AEs (Table 16).

## 5.4 WHEN DO AUGMENTATIONS FAIL TO IMPROVE TRANSFERABILITY?

While augmentation methods mostly increased transferability, in some cases they were counterproductive. Particularly, NeuTrans and Sharp decreased transferability when composed with certain methods. We conducted simple experiments as a preliminary assessment of two conjectures we had concerning when augmentations may harm transferability.

First, we expected augmentation methods that harm model accuracy on benign samples to be less conducive for transferability. As DNNs do not generalize well to benign samples produced by these augmentation methods, we anticipated that adversarial perturbations relying on the augmented samples would also have limited generalizability across models. To support the conjecture, we tested the normally trained DNNs' accuracy on benign samples transformed by each augmentation method. As can be seen from Table 6, NeuTrans and Sharp, which often decrease transferability (Section 5.2 and Figure 2), harmed the DNN accuracy the most (6.5%–58.7% lower accuracy than other methods), supporting our conjecture.

| Augmentation | Inc-v3 | Inc-v4 | Res-50 | Res-101 | Res-152 | IncRes-v2 | Avg. |
|---|---|---|---|---|---|---|---|
| None | 96.2 | 97.4 | 94.5 | 96.3 | 95.8 | 99.8 | 96.7 |
| CS | 94.0 | 95.7 | 95.3 | 94.6 | 95.4 | 99.5 | 95.8 |
| fPCA | 91.6 | 96.8 | 89.9 | 92.8 | 93.6 | 99.4 | 94.0 |
| CJ | 90.0 | 92.3 | 90.3 | 90.3 | 91.4 | 96.8 | 91.9 |
| Admix | 86.7 | 91.6 | 86.8 | 88.9 | 89.7 | 94.6 | 89.7 |
| CutOut | 86.5 | 89.2 | 85.7 | 87.2 | 88.6 | 92.3 | 88.2 |
| GS | 86.6 | 90.3 | 84.7 | 87.6 | 86.5 | 92.7 | 88.1 |
| AutoAugment | 82.9 | 86.2 | 82.1 | 84.3 | 84.4 | 89.8 | 85.0 |
| Sharp | 69.5 | 87.3 | 71.5 | 76.7 | 75.5 | 90.6 | 78.5 |
| NeuTrans | 24.4 | 25.4 | 24.2 | 27.0 | 24.0 | 32.7 | 26.3 |

Table 6: Benign accuracy (%) after applying data augmentation methods. Rows are sorted in a descending order of average transferability.

| Augmentation | Inc-v4 | Res-50 | Res-101 | Res-152 | IncRes-v2 | Avg. |
|---|---|---|---|---|---|---|
| CutOut | 0.568 | 0.583 | 0.581 | 0.574 | 0.591 | 0.579 |
| CS | 0.565 | 0.578 | 0.576 | 0.570 | 0.590 | 0.576 |
| None | 0.564 | 0.575 | 0.573 | 0.568 | 0.586 | 0.573 |
| Admix | 0.563 | 0.575 | 0.573 | 0.567 | 0.586 | 0.573 |
| CJ | 0.560 | 0.575 | 0.573 | 0.568 | 0.584 | 0.572 |
| GS | 0.559 | 0.572 | 0.569 | 0.563 | 0.582 | 0.569 |
| AutoAugment | 0.558 | 0.569 | 0.567 | 0.562 | 0.579 | 0.567 |
| fPCA | 0.560 | 0.568 | 0.566 | 0.561 | 0.578 | 0.567 |
| NeuTrans | 0.546 | 0.556 | 0.554 | 0.549 | 0.565 | 0.554 |
| Sharp | 0.548 | 0.548 | 0.545 | 0.540 | 0.558 | 0.548 |

Table 7: Cosine similarities between gradients of benign images computed on Inc-v3 after applying augmentation methods composed with DST-MI-FGSM, and gradients of other normally trained models on benign images. Rows are sorted in a descending order of average cosine similarity.

Prior work demonstrated that gradient alignment between surrogates and targets is needed for transferability (Demontis et al., 2019). Thus, we expected augmentation methods that estimate target model gradients more accurately to increase transferability further. To assess this conjecture, we evaluated the cosine similarity between the gradients of the Inc-v3 model while using augmentations composed with DST applied to benign samples, and the gradients of other normally trained models on (untransformed) benign samples. The results (Table 7) show some support to the conjecture— NeuTrans and Sharp led to lower cosine similarities with target models' gradients. Yet, the differences in cosine similarities between augmentation methods were small ($\leq 0.031$, on avg.).

## 6    Conclusion and Future Work

Our study uncovered a mostly monotonic relationship between data-augmentation methods and transferability, and helped us identify a simple yet effective composition of data-augmentation methods, ULTIMATECOMBO, that outperforms previously proposed methods when integrated into attacks. The resulting attack should be considered as a standard baseline in follow-up work on transferability. Our work also puts forward conjectures for when augmentation techniques are expected to improve transferability, and offers some empirical support.

In the future, it would be informative to develop a theory that formally explains why augmentation methods help increase transferability. Furthermore, instead of relying on existing augmentation methods originally developed to improve DNN generalizability, an intriguing research direction would be to develop augmentation techniques tailored specifically for improving transferability. Lastly, in addition for assessing the vulnerability of ML models in black-box settings, it would be interesting to evaluate whether the ULTIMATECOMBO-based attack advances methods leveraging AEs for defensive purposes, by deceiving adversaries (e.g., to attain privacy (Cherepanova et al., 2021; Shetty et al., 2018)).

## REPRODUCIBILITY STATEMENT

In the interest of reproducibility, we make our code publicly available at the following repository: `https://tinyurl.com/UltimateComboICLR`.

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

## A  PARALLEL AND SERIAL COMPOSITIONS OF AUGMENTATIONS

Figure 1 illustrates how parallel and serial compositions work.

## B  DNNs USED IN THE EXPERIMENTS

We tested transferability using ten ImageNet DNNs and six CIFAR-10 DNNs. Of the ten ImageNet models, six were normally trained, while others were adversarially trained. Specifically, for normally trained models, we selected: Inception-v3 (Inc-v3) (Szegedy et al., 2016); Inception-v4 (Inc-v4); Inception-ResNet-v2 (IncRes-v2) (Szegedy et al., 2017)); ResNet-v2-50 (Res-50); ResNet-v2-101 (Res-101); and ResNet-v2-152 (Res-152) (He et al., 2016). For adversarially trained models, we selected: Inception-v3-adv (Inc-v3$_{adv}$) (Kurakin et al., 2017a); ens3-Inception-v3 (Inc-v3$_{ens3}$); ens4-Inception-v3 (Inc-v3$_{ens4}$); and ens-adv-Inception-ResNet-v2 (IncRes-v2$_{ens}$) (Tramèr et al., 2018). We obtained the models' PyTorch implementations and weights from a public GitHub repository.[2] All six CIFAR-10 DNNs were normally trained. For this dataset, we used pretrained VGG-11 (VGG) (Simonyan & Zisserman, 2015)), ResNet-50 (Res) (He et al., 2016), DenseNet-121 (DenseNet) (Huang et al., 2017), MobileNet-v2 (MobileNet) (Sandler et al., 2018), GoogleNet (Szegedy et al., 2015), and an Inception-v3 (Inc) DNNs (Szegedy et al., 2016), also implemented in PyTorch.[3]

---

[2]https://github.com/ylhz/tf_to_pytorch_model
[3]https://github.com/huyvnphan/PyTorch_CIFAR10

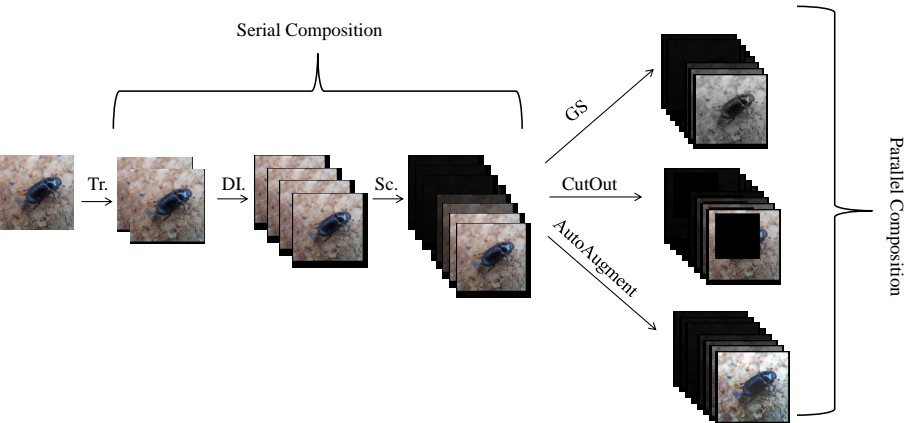

Figure 1: An illustration of serial and parallel compositions. When serially composing augmentations, each augmentation method operates on the output of the previous one. By contrast, in parallel composition, each augmentation method operates independently on the input (or set of inputs). The number of samples grows exponentially in serial composition, whereas it grows linearly in parallel composition. We use serial composition when composing diverse inputs (DI), scaling (Sc.), and translations (Tr.). Other augmentation methods are composed in parallel.

## C  ATTACK AND AUGMENTATION METHOD PARAMETERS

Similarly to Wang et al. (2021b), we set the MI-FGSM decay factor $\mu$=1.0, and the number of iterations $T$=10.

We mostly used default or commonly used parameters of augmentation methods. For CJ, we performed random adjustments of image hue $\in [-0.5, 0.5]$, contrast $\in [0.5, 1.5]$, saturation $\in [0.5, 1.5]$, and brightness$\in [0.5, 1.5]$. For CutOut, we replaced values in selected regions with zeros, and the portion of masked areas compared to image dimensions lied in $[0.02, 0.4]$, with aspect ratios $\in [0.4, 2.5]$. In comparison, for RE, the dimension of masked areas relatively to the image dimensions lied in $[0.02, 0.2]$, with aspect ratios $\in [0.3, 3.3]$. For Sharp, we used the following edge-enhancement mask:

$$\begin{bmatrix} -0.5 & -0.5 & -0.5 \\ -0.5 & 5.0 & -0.5 \\ -0.5 & -0.5 & -0.5 \end{bmatrix}.$$

For diverse inputs, images were transformed with probability 0.5. For the Admix operation, consistently with Wang et al. (2021a), we randomly sampled three images from other categories for mixing as part of the Admix-DT-MI-FGSM attack. However, for the interest of computational efficiency, we use only one image for mixing when composing Admix with other augmentation methods. We did not find that mixing with fewer images harmed performance. In fact, it even improved transferability in some cases. Finally, in CutMix, we picked the top left coordinate $(r_x, r_y)$, the width, $r_w$, and height, $r_h$, of the region to be replaced, using the formulas:

$$r_x \sim \mathcal{U}(0, W), \quad r_w = W\sqrt{1-\lambda},$$

$$r_y \sim \mathcal{U}(0, H), \quad r_h = H\sqrt{1-\lambda},$$

where $\mathcal{U}$ is the uniform distribution, $W$ is the image width, $H$ is the image height, and $\lambda$ is a parameter set to 0.5.

In an attempt to enhance transferability further, we optimized the parameters of a few augmentation methods we considered via grid search. Except for the Gaussian kernel's size used in translation-invariant attacks (Dong et al., 2019), we found that the selected parameters had little impact on

transferability. Specifically, for translations, after considering Gaussian kernels of sizes $\in \{5 \times 5, 7 \times 7, 9 \times 9\}$, we set the default to $7 \times 7$, except for Admix-DT-MI-FGSM, for which the $9 \times 9$ kernel performed best. Table 8 shows that our choice of Admix parameters ($m$=1 and Gaussian kernel size of $9 \times 9$) improves its performance. For GS, we found $\omega_R$, $\omega_G$, and $\omega_B$ had little impact on transferability, as long as the weight assigned to each channel was >0.1. Accordingly, we set $\omega_R$, $\omega_G$, and $\omega_B$ to 0.299, 0.587, and 0.114, respectively, per commonly used values (e.g., in the Python `PyTorch` package[4]). Finally, for CS, we only swapped the blue and green channels, as this led to a minor improvement compared to swapping all three channels.

| Attack | Inc-v3 | Inc-v4 | Res-50 | Res-101 | Res-152 | IncRes-v2 |
|---|---|---|---|---|---|---|
| Admix-DT-MI-FGSM (original) | 99.5 | 92.3 | 88.5 | 87.0 | 85.3 | 90.9 |
| Admix-DT-MI-FGSM (ours) | **100.0** | **94.7** | **91.9** | **90.3** | **88.7** | **93.3** |

Table 8: Transferability rates (%) of AEs crafted via Admix-DT-MI-FGSM against an Inc-v3 surrogate. Our variant sets $m$=1 and the translation's Gaussian kernel to $7 \times 7$ and include original images when calculating gradients, whereas the original work uses $m$=3 and a $9 \times 9$ kernel.

Finally, we clarify that each of our attack combinations emits the original image once, alongside the transformed images. Moreover, when aggregating the gradients, the gradients of the original and transformed images are assigned equal weights. We tested whether weighting the gradients differently (e.g., assigning higher or lower weight to the original sample) can help improve transferability using the GS method. However, we found that equal weights attained the best results.

## D    ATTACKS CONSIDERED BUT EXCLUDED

Besides the three state-of-the-art baselines we experimented with, we considered including two other attacks in the evaluation. Wu et al.'s (2021) attack uses a neural network to create adversarial perturbations robust against transformations for enhanced transferability, and achieves competitive transferability rates. However, unfortunately, we were unable to find a publicly available implementation of the attack. Huang et al.'s (2019) intermediate level attack improve AE transferability by reducing the variance of intermediate activations. We used the official implementation[5] to test the attack on CIFAR-10 with $\epsilon$=0.02 and the VGG or DenseNet models as surrogates. The results showed that the transferability rates were much less competitive that the three baselines we considered (50.34% vs. >54.00% average transferability with a VGG surrogate, and 45.68% vs. >56.56% average transferability with a DenseNet surrogate). Therefore, we removed the intermediate level attack for the remaining experiments.

## E    INDIVIDUAL AUGMENTATIONS

Table 9 presents the transferability rates when integrating individual augmentation methods into MI-FGSM. Table 10 presents the transferability when composing individual augmentation methods with DST. Trasnferability rates were computed on ImageNet, using the Inc-v3 DNN as a surrogate and the other normally trained DNNs as victims ($\epsilon = \frac{16}{255}$). Notice how composing color-space augmentations (specifically, CS, CJ, and GS) with DST helps improve transferability over the baselines (Table 10).

## F    TRANSFERABILITY RATES ON IMAGENET

Tables 11 and 12 detail the trasferability rates on ImageNet, from all ten DNNs to normally and adversarially trained models, respectively. Here, we also consider transferring AEs from adversarially trained surrogates. Table 13 shows the transferability rates on ImageNet from Inc-v3 to other normally traiend models with varied perturbation norms (i.e., values of $\epsilon$).

---

[4] `https://bit.ly/3ynCyUD`
[5] `https://github.com/CUAI/Intermediate-Level-Attack`

## G    TRANSFERABILITY RATES ON CIFAR-10

Tables 14 and 15 report attack tranferability rates from all six normally trained DNNs to all other victim DNNs for $\epsilon$=0.02 and $\epsilon$=0.04, respectively.

## H    THE MONOTONICITY OF TRANSFERABILITY WHEN ADDING AUGMENTATIONS

Figure 2 depicts a visual summary of the experiment presented in Section 5.2, demonstrating how the relationship between augmentation methods and transferability is mostly monotonic.

## I    ATTACK RUN-TIME

MI-FGSM's time complexity is predominated by the gradient computation steps. Accordingly, the attacks' run-times are directly affected by the number of samples the augmentation methods create (i.e., samples emitted by $D(\cdot)$ in Algorithm 1): The more samples emitted by the augmentation method, the more back-propagation would be required to compute gradients for updating the adversarial examples in each iteration, thus increasing the AE-generation time. The empirical measurements corroborate this intuition (Table 16). Overall, we can see that DST augments MI-FGSM with the least samples, leading to the fastest attack (DST-MI-FGSM). GS-DST-MI-FGSM is the second fastest attack, while ULTIMATECOMBO is slower than Admix-DT-MI-FGSM but substantially faster than DST-VMI-FGSM. We note that no particular effort was invested to make GS-DST-MI-FGSM and ULTIMATECOMBO more time-efficient (e.g., stacking augmented samples for parallel computation, similarly to Admix-DT-MI-FGSM). Moreover, since transferability-based attacks generate AEs offline, and only once per surrogate model, as long as an attack is not prohibitively slow, attack run-time is a marginal consideration for selecting an attack compared to transferability rates.

## J    DEFENSE PARAMETERS

We used standard parameters when attacking defenses. For RS, we used a normally trained ResNet-50 and set $\sigma$ to 0.25, following Cohen et al. (2019). For ARS, the target model was ResNet-50 trained with isotropic Gaussian-noise augmentations (sampled from $\mathcal{N}(0, 0.25)$), and $\sigma$ was set to 0.25 during prediction, per Salman et al. (2019). In both cases, we used 10,000 noisified samples during inference. For Bit-Red, we used a squeezer with bit-depth of one, in accordance with Xu et al. (2018). Finally, we used the default NRP parameters and pre-trained model from the official GitHub repository (Naseer et al., 2020).

| Attack | Inc-v3 | Inc-v4 | Res-50 | Res-101 | Res-152 | IncRes-v2 |
|---|---|---|---|---|---|---|
| MI-FGSM | 100.0 | 54.7 | 48.9 | 43.5 | 41.1 | 50.6 |
| fPCA-MI-FGSM | 100.0 | 70.7 | 65.4 | 59.6 | 57.6 | 69.2 |
| CS-MI-FGSM | 100.0 | 57.5 | 54.7 | 49.9 | 46.3 | 56.7 |
| CJ-MI-FGSM | 100.0 | 66.4 | 61.8 | 57.5 | 54.7 | 65.4 |
| GS-MI-FGSM | 100.0 | 62.9 | 61.4 | 56.5 | 51.8 | 62.4 |
| RE-MI-FGSM | 100.0 | 55.1 | 52.1 | 46.6 | 43.9 | 52.2 |
| CutMix-MI-FGSM | 63.0 | 34.7 | 33.9 | 30.0 | 33.1 | 31.2 |
| CutOut-MI-FGSM | 100.0 | 58.2 | 54.4 | 49.6 | 45.7 | 55.9 |
| NeuTrans-MI-FGSM | 96.4 | 44.4 | 39.5 | 35.6 | 33.6 | 38.2 |
| Sharp-MI-FGSM | 99.3 | 44.7 | 41.0 | 34.3 | 34.4 | 40.2 |
| AutoAugment-MI-FGSM | 100.0 | 61.1 | 56.6 | 50.1 | 48.5 | 58.4 |

Table 9: Transferability rates (%) on ImageNet from a normally trained Inc-v3 surrogate to normally trained target models (columns) when integrating individual augmentation methods into MI-FGSM-based attacks.

| Attack | Inc-v3 | Inc-v4 | Res-50 | Res-101 | Res-152 | IncRes-v2 |
|---|---|---|---|---|---|---|
| Admix-DT-MI-FGSM | 99.5 | 92.3 | 88.5 | 87.0 | 85.3 | 90.9 |
| DST-MI-FGSM | 100.0 | 92.9 | 89.5 | 87.2 | 86.4 | 91.2 |
| DST-VMI-FGSM | 100.0 | 94.7 | 90.7 | 88.9 | 89.1 | 92.6 |
| fPCA-DST-MI-FGSM | 100.0 | 94.3 | 90.3 | 88.6 | 87.3 | 90.8 |
| CS-DST-MI-FGSM | 100.0 | 94.7 | 92.8 | 90.3 | 88.7 | 93.3 |
| CJ-DST-MI-FGSM | 100.0 | 94.9 | 92.3 | 90.9 | 90.2 | 94.1 |
| GS-DST-MI-FGSM | 100.0 | 95.6 | 93.7 | 91.8 | 90.9 | 94.9 |
| RE-DST-MI-FGSM | 100.0 | 95.3 | 91.5 | 89.7 | 88.2 | 93.5 |
| CutMix-DST-MI-FGSM | 69.5 | 61.8 | 59.6 | 58.1 | 56.4 | 58.6 |
| CutOut-DST-MI-FGSM | 100.0 | 95.0 | 91.4 | 89.8 | 89.1 | 93.4 |
| NeuTrans-DST-MI-FGSM | 97.8 | 87.2 | 82.5 | 79.8 | 80.7 | 84.4 |
| Sharp-DST-MI-FGSM | 99.9 | 95.6 | 91.3 | 89.6 | 89.4 | 92.9 |
| AutoAugment-DST-MI-FGSM | 100.0 | 94.1 | 91.1 | 88.6 | 87.5 | 92.9 |

Table 10: Transferability rates (%) on ImageNet from a normally trained Inc-v3 surrogate to normally trained target models (columns) when integrating individual augmentation methods composed with DST into MI-FGSM-based attacks. Admix-DT-MI-FGSM, DST-MI-FGSM, and DST-VMI-FGSM are baseline attacks from prior work.

| Model | Attack | Inc-v3 | Inc-v4 | Res-50 | Res-101 | Res-152 | IncRes-v2 |
|---|---|---|---|---|---|---|---|
| Inc-v3 | Admix-DT-MI-FGSM | *99.5* | 92.3 | 88.5 | 87.0 | 85.3 | 90.9 |
| | DST-MI-FGSM | **100.0** | 92.9 | 89.5 | 87.2 | 86.4 | 91.2 |
| | DST-VMI-FGSM | **100.0** | 94.7 | 90.7 | 88.9 | 89.1 | 92.6 |
| | GS-DST-MI-FGSM | **100.0** | *95.6* | *93.7* | *91.8* | *90.9* | *94.9* |
| | UltimateCombo | **100.0** | **98.0** | **95.1** | **94.3** | **92.7** | **97.1** |
| Inc-v4 | Admix-DT-MI-FGSM | 93.7 | 99.3 | 86.7 | 84.9 | 84.7 | 89.6 |
| | DST-MI-FGSM | 94.4 | **100.0** | 90.2 | 88.1 | 88.2 | 92.8 |
| | DST-VMI-FGSM | 95.3 | 99.9 | 91.0 | 89.9 | 88.4 | 93.5 |
| | GS-DST-MI-FGSM | *96.5* | **100.0** | *94.1* | *92.5* | *93.0* | *95.4* |
| | UltimateCombo | **98.1** | 99.9 | **94.8** | **95.0** | **94.6** | **96.8** |
| Res-101 | Admix-DT-MI-FGSM | 82.6 | 78.1 | 93.5 | 97.4 | 93.9 | 79.3 |
| | DST-MI-FGSM | 86.7 | 83.2 | 97.3 | **99.9** | 96.6 | 84.9 |
| | DST-VMI-FGSM | 88.3 | *85.0* | 97.6 | **99.9** | 96.9 | 87.2 |
| | GS-DST-MI-FGSM | *89.0* | 84.8 | 97.6 | *99.8* | *97.7* | *87.6* |
| | UltimateCombo | **93.0** | **90.4** | **98.1** | 99.7 | **97.8** | **91.8** |
| IncRes-v2 | Admix-DT-MI-FGSM | 93.8 | 91.9 | 91.1 | 90.6 | 89.5 | 98.9 |
| | DST-MI-FGSM | 95.8 | 94.2 | 93.5 | 92.0 | 92.4 | *99.8* |
| | DST-VMI-FGSM | 95.8 | 94.7 | 94.0 | 92.9 | 92.9 | 99.7 |
| | GS-DST-MI-FGSM | *96.5* | *95.6* | *95.5* | *94.2* | *94.7* | **100.0** |
| | UltimateCombo | **98.2** | **97.1** | **96.3** | **96.5** | **95.7** | **100.0** |
| Res-50 | Admix-DT-MI-FGSM | 83.9 | 80.0 | 97.8 | 93.4 | 93.2 | 78.9 |
| | DST-MI-FGSM | 89.1 | 84.1 | *99.9* | 97.3 | 96.8 | 84.3 |
| | DST-VMI-FGSM | 88.7 | 84.7 | **100.0** | 98.1 | 97.1 | 86.7 |
| | GS-DST-MI-FGSM | *90.3* | *85.7* | 99.9 | 98.3 | 97.9 | 87.5 |
| | UltimateCombo | **94.1** | **91.6** | 99.9 | **99.0** | **98.2** | **90.9** |
| Res-152 | Admix-DT-MI-FGSM | 82.9 | 79.0 | 92.5 | 92.3 | 96.5 | 77.9 |
| | DST-MI-FGSM | 86.7 | 83.6 | 95.5 | 97.1 | **99.8** | 83.1 |
| | DST-VMI-FGSM | *88.4* | *85.3* | 95.6 | 97.2 | *99.7* | *85.6* |
| | GS-DST-MI-FGSM | 87.7 | 84.3 | *96.7* | **97.6** | **99.8** | 85.3 |
| | UltimateCombo | **91.8** | **90.4** | **97.0** | *97.3* | 99.5 | **90.3** |
| Inc-v3$_{adv}$ | Admix-DT-MI-FGSM | 92.5 | 87.9 | 89.0 | 88.2 | 86.8 | 88.3 |
| | DST-MI-FGSM | 93.6 | 90.9 | 91.2 | 90.6 | 89.5 | 91.7 |
| | DST-VMI-FGSM | *95.9* | 91.7 | 91.3 | 91.7 | 90.3 | 92.6 |
| | GS-DST-MI-FGSM | 95.2 | *92.8* | *93.8* | *92.8* | *92.2* | *94.2* |
| | UltimateCombo | **97.1** | **95.5** | **95.3** | **94.6** | **94.0** | **95.8** |
| Inc-v3$_{ens3}$ | Admix-DT-MI-FGSM | 87.9 | 83.4 | 85.3 | 84.5 | 83.9 | 85.3 |
| | DST-MI-FGSM | 90.2 | 85.9 | 86.7 | 86.0 | 85.3 | 87.3 |
| | DST-VMI-FGSM | 90.4 | 87.1 | 88.9 | 88.0 | 87.1 | 88.7 |
| | GS-DST-MI-FGSM | *93.2* | *90.4* | *90.7* | *90.0* | *90.1* | *90.7* |
| | UltimateCombo | **94.5** | **92.1** | **93.2** | **92.3** | **92.1** | **92.9** |
| Inc-v3$_{ens4}$ | Admix-DT-MI-FGSM | 86.0 | 80.3 | 81.4 | 82.3 | 81.1 | 80.7 |
| | DST-MI-FGSM | 88.8 | 85.6 | 83.4 | 85.9 | 83.7 | 84.4 |
| | DST-VMI-FGSM | 89.8 | 84.8 | 85.6 | 86.2 | 85.2 | 85.3 |
| | GS-DST-MI-FGSM | *92.8* | *88.4* | *90.2* | *89.5* | *87.8* | *88.5* |
| | UltimateCombo | **94.5** | **90.2** | **92.0** | **91.0** | **91.1** | **91.5** |
| IncRes-v2$_{ens}$ | Admix-DT-MI-FGSM | 82.8 | 80.4 | 81.8 | 79.5 | 80.3 | 83.5 |
| | DST-MI-FGSM | 85.5 | 82.9 | 84.8 | 84.5 | 83.9 | 88.1 |
| | DST-VMI-FGSM | 87.2 | 84.3 | 85.3 | 85.7 | 85.6 | 88.6 |
| | GS-DST-MI-FGSM | *90.1* | *88.2* | **91.0** | *89.7* | *88.6* | *91.6* |
| | UltimateCombo | **91.7** | **90.9** | *90.9* | **90.9** | **89.7** | **93.3** |

Table 11: Transferability rates (%) on ImageNet from ten surrogates (rows) to normally trained target models (columns). All attacks are black-box, except for when the surrogate and target models are the same.

| Model | Attack | Inc-v3$_{adv}$ | Inc-v3$_{ens3}$ | Inc-v3$_{ens4}$ | IncRes-v2$_{ens}$ |
|---|---|---|---|---|---|
| Inc-v3 | Admix-DT-MI-FGSM | 84.6 | 84.3 | 83.5 | 70.8 |
| | DST-MI-FGSM | 81.3 | 81.2 | 77.7 | 61.4 |
| | DST-VMI-FGSM | 84.4 | 84.8 | 82.9 | 69.2 |
| | GS-DST-MI-FGSM | *87.3* | *88.5* | *85.5* | *72.2* |
| | ULTIMATECOMBO | **88.2** | **88.7** | **86.7** | **72.6** |
| Inc-v4 | Admix-DT-MI-FGSM | 82.7 | 83.3 | 81.3 | 73.7 |
| | DST-MI-FGSM | 80.6 | 81.8 | 80.8 | 70.5 |
| | DST-VMI-FGSM | 84.3 | 86.0 | 83.0 | 74.6 |
| | GS-DST-MI-FGSM | *87.6* | **89.5** | *87.2* | *78.0* |
| | ULTIMATECOMBO | **88.6** | *89.4* | **88.4** | **78.2** |
| Res-101 | Admix-DT-MI-FGSM | 79.5 | 80.4 | 78.6 | 71.2 |
| | DST-MI-FGSM | 78.9 | 78.7 | 76.7 | 68.7 |
| | DST-VMI-FGSM | 82.0 | 83.0 | 80.9 | 72.5 |
| | GS-DST-MI-FGSM | *82.3* | *83.7* | *81.1* | *73.2* |
| | ULTIMATECOMBO | **83.5** | **86.7** | **82.8** | **76.8** |
| IncRes-v2 | Admix-DT-MI-FGSM | 89.0 | 89.0 | 88.7 | 87.1 |
| | DST-MI-FGSM | 87.2 | 89.2 | 86.4 | 82.9 |
| | DST-VMI-FGSM | 88.8 | 90.6 | 87.8 | 85.5 |
| | GS-DST-MI-FGSM | *91.1* | *92.2* | *90.0* | *88.2* |
| | ULTIMATECOMBO | **92.2** | **92.6** | **92.0** | **88.5** |
| Res-50 | Admix-DT-MI-FGSM | 79.1 | 78.0 | 77.1 | 68.0 |
| | DST-MI-FGSM | 76.1 | 77.7 | 75.6 | 63.6 |
| | DST-VMI-FGSM | 80.7 | 80.7 | *79.5* | 69.7 |
| | GS-DST-MI-FGSM | *81.4* | *83.6* | 78.9 | *70.4* |
| | ULTIMATECOMBO | **85.1** | **85.7** | **83.4** | **74.0** |
| Res-152 | Admix-DT-MI-FGSM | 77.7 | 78.2 | 75.9 | 71.9 |
| | DST-MI-FGSM | 75.0 | 78.4 | 75.7 | 70.1 |
| | DST-VMI-FGSM | 79.1 | 81.0 | *79.5* | 72.7 |
| | GS-DST-MI-FGSM | *79.6* | *82.1* | 79.0 | 71.3 |
| | ULTIMATECOMBO | **83.6** | **85.0** | **82.5** | **74.8** |
| Inc-v3$_{adv}$ | Admix-DT-MI-FGSM | 98.3 | 93.0 | 92.2 | 87.4 |
| | DST-MI-FGSM | *99.7* | 93.0 | 93.5 | 85.4 |
| | DST-VMI-FGSM | 99.6 | 94.1 | 93.7 | 88.1 |
| | GS-DST-MI-FGSM | **99.9** | *95.1* | *94.1* | *88.5* |
| | ULTIMATECOMBO | **99.9** | **96.1** | **96.2** | **91.0** |
| Inc-v3$_{ens3}$ | Admix-DT-MI-FGSM | 88.3 | 98.2 | 91.6 | 84.6 |
| | DST-MI-FGSM | 87.8 | **99.9** | 91.9 | 83.0 |
| | DST-VMI-FGSM | 90.3 | *99.7* | 92.6 | 84.8 |
| | GS-DST-MI-FGSM | *91.6* | **99.9** | *93.4* | *85.8* |
| | ULTIMATECOMBO | **94.0** | **99.9** | **94.7** | **89.6** |
| Inc-v4$_{ens4}$ | Admix-DT-MI-FGSM | 88.3 | 90.3 | 98.7 | 83.4 |
| | DST-MI-FGSM | 88.2 | 91.8 | 99.8 | 83.8 |
| | DST-VMI-FGSM | 89.7 | 91.7 | **100.0** | 85.6 |
| | GS-DST-MI-FGSM | *91.8* | *94.1* | *99.9* | *86.9* |
| | ULTIMATECOMBO | **93.4** | **95.4** | *99.9* | **90.2** |
| IncRes-v2$_{ens}$ | Admix-DT-MI-FGSM | 86.2 | 87.7 | 88.6 | 96.3 |
| | DST-MI-FGSM | 88.2 | 89.4 | 89.8 | 98.7 |
| | DST-VMI-FGSM | 89.4 | 90.5 | 91.3 | **99.3** |
| | GS-DST-MI-FGSM | *92.0* | *93.4* | *92.7* | *99.1* |
| | ULTIMATECOMBO | **92.5** | **93.5** | **93.1** | **99.3** |

Table 12: Transferability rates on ImageNet (%) from ten surrogates (rows) to adversarially trained target models (columns).

| Epsilon | Attack | Inc-v3 | Inc-v4 | Res-152 | IncRes-v2 | Res-50 | Res-101 |
|---------|--------|--------|--------|---------|-----------|--------|---------|
| 8/255 | Admix-DT-MI-FGSM | 98.3 | 67.8 | 50.0 | 59.1 | 57.4 | 53.5 |
| | DST-MI-FGSM | 99.7 | 77.1 | 62.9 | 70.9 | 69.8 | 64.8 |
| | DST-VMI-FGSM | 99.5 | 77.9 | 64.1 | 72.9 | 72.2 | 66.7 |
| | GS-DST-MI-FGSM | *99.5* | *81.3* | *71.1* | *77.6* | *77.3* | *72.0* |
| | UltimateCombo | **99.7** | **86.0** | **75.0** | **81.8** | **81.2** | **76.3** |
| 24/255 | Admix-DT-MI-FGSM | 99.8 | 94.9 | 88.2 | 93.1 | 89.4 | 88.4 |
| | DST-MI-FGSM | 100.0 | 97.8 | 93.6 | 96.8 | 95.1 | 93.9 |
| | DST-VMI-FGSM | 100.0 | 97.9 | 95.0 | 97.1 | 96.2 | 95.6 |
| | GS-DST-MI-FGSM | *100.0* | *98.5* | *96.7* | *98.7* | *97.0* | *97.0* |
| | UltimateCombo | **100.0** | **99.3** | **97.4** | **99.2** | **97.5** | **98.4** |

Table 13: Transferability rates (%) on ImageNet, from a Inc-v3 surrogate to other normally trained models, with perturbation norms $\epsilon \in \{\frac{8}{255}, \frac{24}{255}\}$ other than the default $\epsilon = \frac{16}{255}$.

| Model | Attack | VGG | Res | DenseNet | MobileNet | GoogleNet | Inc |
|-------|--------|-----|-----|----------|-----------|-----------|-----|
| VGG | Admix-DT-MI-FGSM | 92.1 | 67.8 | 66.6 | 78.8 | 69.9 | 67.5 |
| | DST-MI-FGSM | 93.5 | 73.5 | 71.5 | 80.4 | 71.6 | 71.5 |
| | DST-VMI-FGSM | 68.9 | 52.3 | 50.5 | 61.6 | 54.0 | 51.6 |
| | GS-DST-MI-FGSM | *94.5* | *76.5* | *75.7* | *85.8* | *78.3* | *77.3* |
| | UltimateCombo | **95.0** | **77.0** | **77.7** | **86.5** | **80.2** | **77.9** |
| Res | Admix-DT-MI-FGSM | 47.1 | 94.9 | 74.2 | 78.9 | 68.1 | 64.3 |
| | DST-MI-FGSM | 52.4 | 98.5 | 83.1 | 86.5 | 75.9 | 71.5 |
| | DST-VMI-FGSM | 40.9 | 75.2 | 59.2 | 64.7 | 56.1 | 54.4 |
| | GS-DST-MI-FGSM | *58.7* | *98.4* | *86.9* | *90.3* | *80.3* | *78.6* |
| | UltimateCombo | **60.8** | **98.3** | **88.5** | **91.0** | **80.7** | **80.6** |
| DenseNet | Admix-DT-MI-FGSM | 50.8 | 78.4 | 94.7 | 81.4 | 70.8 | 70.7 |
| | DST-MI-FGSM | 56.3 | 86.9 | 98.5 | 86.8 | 76.5 | 77.4 |
| | DST-VMI-FGSM | 40.6 | 62.2 | 77.6 | 67.1 | 58.6 | 54.3 |
| | GS-DST-MI-FGSM | *61.5* | *88.8* | *98.4* | *89.1* | *81.1* | *81.8* |
| | UltimateCombo | **64.3** | **90.0** | **98.6** | **92.2** | **82.4** | **84.2** |
| MobileNet | Admix-DT-MI-FGSM | 34.6 | 54.5 | 50.8 | 99.8 | 70.3 | 64.0 |
| | DST-MI-FGSM | 37.3 | 60.0 | 54.4 | 100.0 | 76.8 | 71.3 |
| | DST-VMI-FGSM | 32.6 | 50.7 | 46.5 | 90.6 | 64.8 | 59.3 |
| | GS-DST-MI-FGSM | *42.0* | *63.6* | *62.2* | *100.0* | *83.1* | *78.7* |
| | UltimateCombo | **44.2** | **67.9** | **62.5** | **100.0** | **86.4** | **82.1** |
| GoogleNet | Admix-DT-MI-FGSM | 42.8 | 63.0 | 59.0 | 88.2 | 99.9 | 78.6 |
| | DST-MI-FGSM | 45.7 | 65.8 | 62.5 | 92.7 | 100.0 | 84.2 |
| | DST-VMI-FGSM | 40.5 | 56.2 | 55.2 | 77.7 | 90.2 | 67.2 |
| | GS-DST-MI-FGSM | *47.7* | *65.3* | *62.5* | *93.4* | *100.0* | *87.0* |
| | UltimateCombo | **49.2** | **67.9** | **65.0** | **94.0** | **100.0** | **90.6** |
| Inc | Admix-DT-MI-FGSM | 46.8 | 65.4 | 63.3 | 89.6 | 85.4 | 98.0 |
| | DST-MI-FGSM | 50.9 | 68.1 | 70.1 | 93.1 | 90.2 | 99.7 |
| | DST-VMI-FGSM | 39.0 | 51.1 | 48.6 | 64.1 | 59.0 | 74.5 |
| | GS-DST-MI-FGSM | *50.8* | *69.0* | *68.0* | *92.6* | *89.7* | *99.6* |
| | UltimateCombo | **52.6** | **72.2** | **70.0** | **95.3** | **93.0** | **99.8** |

Table 14: Transferability rates (%) on CIFAR-10, from normally trained surrogates (rows) to normally trained target models (columns), with a perturbation norm $\epsilon$=0.02.

| Model | Attack | VGG | Res | DenseNet | MobileNet | GoogleNet | Inc |
|---|---|---|---|---|---|---|---|
| VGG | Admix-DT-MI-FGSM | 97.4 | 91.0 | 90.1 | 93.2 | 89.2 | 88.8 |
| | DST-MI-FGSM | 97.5 | 91.7 | 92.9 | 94.2 | 89.8 | 89.7 |
| | DST-VMI-FGSM | 90.4 | 75.5 | 72.0 | 79.3 | 75.6 | 74.4 |
| | GS-DST-MI-FGSM | *97.9* | *95.1* | *94.8* | *96.1* | *92.9* | *92.4* |
| | UltimateCombo | **98.2** | **95.2** | **95.5** | **96.8** | **94.6** | **92.5** |
| Res | Admix-DT-MI-FGSM | 69.6 | 99.4 | 92.8 | 92.8 | 85.3 | 85.5 |
| | DST-MI-FGSM | 77.5 | 100.0 | 96.9 | 95.9 | 91.6 | 90.9 |
| | DST-VMI-FGSM | 57.7 | 88.6 | 78.0 | 81.8 | 75.6 | 74.2 |
| | GS-DST-MI-FGSM | *86.1* | *100.0* | *98.2* | *97.9* | *93.8* | *94.5* |
| | UltimateCombo | **88.7** | **100.0** | **99.0** | **98.9** | **95.7** | **96.3** |
| DenseNet | Admix-DT-MI-FGSM | 78.8 | 94.1 | 98.3 | 93.9 | 89.4 | 90.0 |
| | DST-MI-FGSM | 85.5 | 98.0 | 99.8 | 97.5 | 93.5 | 94.7 |
| | DST-VMI-FGSM | 59.8 | 80.2 | 88.2 | 80.8 | 73.0 | 72.5 |
| | GS-DST-MI-FGSM | *90.4* | *97.9* | *99.9* | *98.5* | *95.6* | *96.1* |
| | UltimateCombo | **92.2** | **99.0** | **99.9** | **99.1** | **95.7** | **97.1** |
| MobileNet | Admix-DT-MI-FGSM | 51.5 | 76.5 | 70.6 | 99.9 | 87.0 | 86.1 |
| | DST-MI-FGSM | 59.6 | 82.4 | 78.8 | 100.0 | 93.3 | 91.2 |
| | DST-VMI-FGSM | 48.7 | 68.3 | 63.6 | 94.2 | 78.4 | 76.8 |
| | GS-DST-MI-FGSM | *70.1* | *87.6* | *86.9* | *100.0* | *95.2* | *93.8* |
| | UltimateCombo | **71.1** | **90.0** | **88.2** | **100.0** | **96.8** | **95.4** |
| GoogleNet | Admix-DT-MI-FGSM | 69.7 | 86.4 | 81.8 | 97.5 | 100.0 | 95.0 |
| | DST-MI-FGSM | 73.1 | 88.6 | 85.1 | 97.8 | 100.0 | 97.1 |
| | DST-VMI-FGSM | 61.8 | 73.8 | 72.8 | 86.8 | 93.1 | 83.3 |
| | GS-DST-MI-FGSM | *77.6* | *89.2* | *88.5* | *99.0* | *100.0* | *98.3* |
| | UltimateCombo | **78.1** | **92.4** | **90.4** | **99.0** | **100.0** | **98.9** |
| Inc | Admix-DT-MI-FGSM | 74.2 | 87.1 | 88.1 | 97.0 | 96.0 | 99.3 |
| | DST-MI-FGSM | 79.5 | 92.2 | 91.2 | 98.9 | 98.4 | 100.0 |
| | DST-VMI-FGSM | 55.4 | 65.3 | 63.0 | 74.7 | 72.2 | 81.2 |
| | GS-DST-MI-FGSM | *79.7* | *90.6* | *91.0* | *97.7* | *98.3* | *100.0* |
| | UltimateCombo | **82.6** | **93.8** | **92.7** | **98.9** | **99.3** | **100.0** |

Table 15: Transferability rates (%) on CIFAR-10, from normally trained surrogates (rows) to normally trained target models (columns), with a perturbation norm $\epsilon$=0.04.

| | Augmented samples | Time (s) |
|---|---|---|
| Admix-DT-MI-FGSM | 15 | 1.68 |
| DST-MI-FGSM | 5 | **0.72** |
| DST-VMI-FGSM | 105 | 11.29 |
| GS-DST-MI-FGSM | 10 | *1.47* |
| UltimateCombo | 30 | 4.63 |

Table 16: The number of samples augmented and the average time of crafting an AE (seconds per images) for different attacks. Times were measured on ImageNet, while attacking an Inc-v3 surrogate, and averaged for 1,000 samples. Experiments were executed on an Nvidia A5000 GPU.

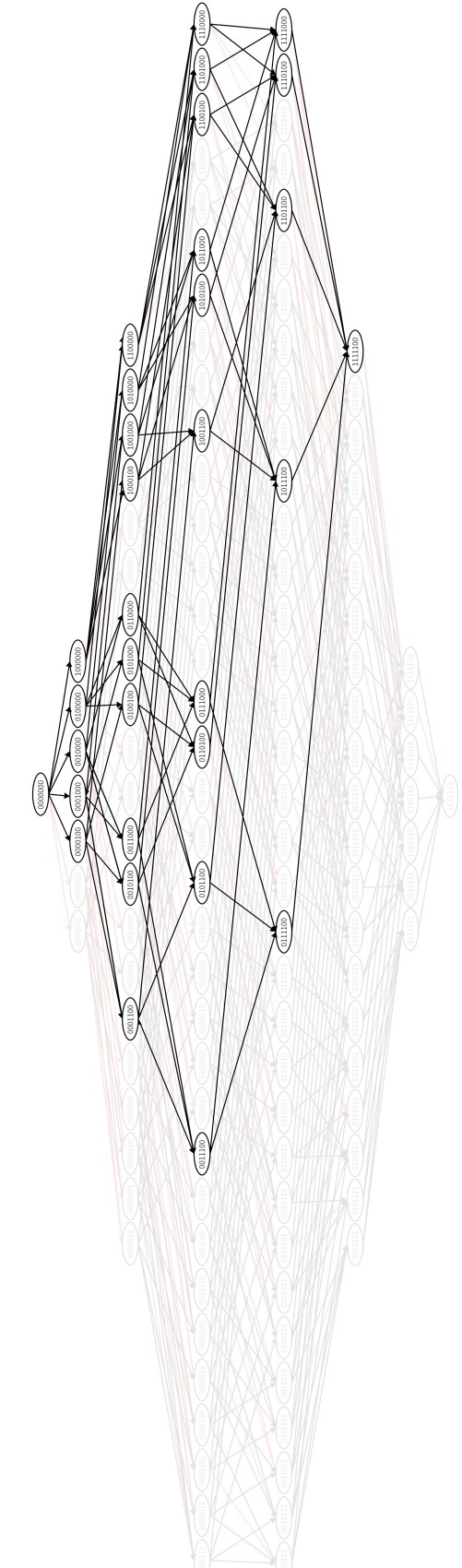

Figure 2: Best viewed after zooming in. Each node represents a composition of augmentation methods. The binary string within a node encodes the composition: Each bit, from the most to the least significant, denotes whether Admix (most significant bit), GS, CutOut, AutoAugment, DST-MI-FGSM, NeuTrans, and Sharp (least significant bit), respectively, is included (1) or excluded (0) from the composition. An edge from node $u$ to $v$ is included if $v$ includes exactly one more augmentation method compared to $u$. As explained in Section 5.2, we computed the average transferability rate per composition on ImageNet, from an Inc-v3 surrogate DNN to the remaining nine models as victim models. An edge $(u, v)$ is colored in black (resp. red) if $v$'s composition achieves higher or equal (resp. lower) average transferability rates than $u$ when integrated into MI-FGSM. We faded away nodes containing NeuTrans or Sharp and their corresponding edges. Notice how all the remaining (unfaded) edges are black, showing that the relationship between the (average) transferability and the remaining augmentations is monotonic (i.e., more augmentations composed → ≥transferability). Said differently, only NeuTrans and Sharp harm transferability in some cases.

