# OpenReview forum: "The Ultimate Combo: Boosting Adversarial Example Transferability by Composing Data Augmentations"
_ICLR.cc/2023/Conference — Submitted to ICLR 2023_

### Official Review · Reviewer_fNhP · 2022-10-21

**Confidence:** 5
**Correctness:** 3
**Technical Novelty And Significance:** 1
**Empirical Novelty And Significance:** 2
**Recommendation:** 3

**Clarity, Quality, Novelty And Reproducibility:**

The paper is easy to follow. The analysis is sound and reasonable. However, a large part of the study is based on established work (see W1). Also, the paper does not compare or discuss their method with existing works (see W2).

**Strength And Weaknesses:**

__Strength__

__[S1]__ The paper is easy to follow.

__[S2]__ The method is simple, practical and effective.

__Weakness__

__[W1] Lack of novelty.__ The paper studies the use of data augmentation in improving the attack transferability of adversarial examples. Apparently, all the data augmentation schemes are based on established works. There are no new theories or methods about data augmentation and attack transferability are introduced.

__[W2] Lack of comparison with related works.__ The paper should discuss and compare the method with existing works, e.g. [Huang et al., 2019], [Wang et al., 2021], [Wu et al., 2021].

__[W2] Insufficient experiments.__ The study is only conducted on the ImageNet-compatible dataset using MI-FGSM on Inception- and ResNet-based models. It is unknown if the findings will be the same on other datasets (e.g. CIFAR-100), attack methods (e.g. I-FGSM) and architectures (e.g. VGG).

__Ref.__

[Huang et al., 2019]: Enhancing Adversarial Example Transferability with an Intermediate Level Attack.

[Wang et al., 2021]: Enhancing the Transferability of Adversarial Attacks through Variance Tuning.

[Wu et al., 2021]: Improving the Transferability of Adversarial Samples with Adversarial Transformations.


**Summary Of The Paper:**

The paper studies the use of various data augmentation in boosting the attack transferability of adversarial examples. Overall, the paper is clear and the experimental results and analysis support the claim.

**Summary Of The Review:**

Overall, this paper proposes a simple way to combine different data augmentation methods to improve the attack transferability of adversarial samples. Due to the lack of novelty and comparison with related works, I tend to vote for rejecting the submission.

---

> ### Author Response · Authors · 2022-11-19
> **Addressing comments and revision changes**
>
> We thank the reviewer for the constructive comments.
>
> > *[W1] Lack of novelty.* The paper studies the use of data augmentation in improving the attack transferability of adversarial examples. Apparently, all the data augmentation schemes are based on established works. There are no new theories or methods about data augmentation and attack transferability are introduced.
>
> Our work is the first to systematically analyze and compare the contribution of a wide range of data-augmentation techniques to transferability. Although our work relies on existing schemes originally proposed to help ML models generalize, we show that by composing these, we can significantly improve the start-of-the-art in transferability. We agree that introducing new theories that can help explain our results or developing data augmentation schemes optimized for transferability are important open problems that result from our work. However, we believe that highlighting the usefulness of our approach and the importance of these questions is a significant first step. In the revised version (Section 6), we added a description of the open questions. We hope our work will motivate further research in these directions.
>
> > *[W2] Lack of comparison with related works.* The paper should discuss and compare the method with existing works, e.g. [Huang et al., 2019], [Wang et al., 2021], [Wu et al., 2021].
>
> We thank the reviewer for pointing out the related work. We added Wang et al.’s (2021) variance-tuning attack (DST-VMI-FGSM) to all our comparisons. While this attack attains better transferability than other baselines (Admix-DT-FGSM and DSG-MI-FGSM) on ImageNet, we found that it is still markedly outperformed by GS-DST-MI-FGSM and UltimateCombo. Wu et al. (2021) did not publish their implementation, so we could not reproduce their results for comparison. Finally, preliminary experiments with Huang et al.’s (2019) attack on CIFAR-10 demonstrated weaker transferability rates than the other baselines we considered, leading us to omit it from further experiments. We discuss Wu et al.’s and Huang et al.’s attacks in Appendix D.
>
> > *[W3] Insufficient experiments.* The study is only conducted on the ImageNet-compatible dataset using MI-FGSM on Inception- and ResNet-based models. It is unknown if the findings will be the same on other datasets (e.g. CIFAR-100), attack methods (e.g. I-FGSM) and architectures (e.g. VGG).
>
> To evidence the generality of our findings, in the revised version, we added new experiments comparing our method to previous work across different datasets, perturbation strengths, and defenses. All experiments support our claim and show significant improvement in transferability compared to previous state-of-the-art methods. The new settings we tested are:
>
> * CIFAR-10 using different architectures (Tables 14 and 15).
> * Different epsilon values with ImageNet (Table 13)
> * Different epsilon values with CIFAR-10 (Tables 14 and 15)
> * Evaluating TRS, a state-of-the-art defense tailored for transferability-based attacks [Yang et al., 2021] (Table 5).
>
>
> ## Refs.
>
> [Yang et al., 2021] Yang et al. “TRS: Transferability reduced ensemble via encouraging gradient diversity and model smoothness.” Proc. NeurIPS, 2021.

---

> > ### Comment · Reviewer_fNhP · 2022-11-28
> > **After rebuttal**
> >
> > Thank you very much for the response. I appreciate the effort that the authors put into addressing my questions. However, I still believe that the novelty of the paper does not reach the bar of ICLR. It is recommended to study more novel augmentation mechanisms for adversarial transferability or new ways to combine the augmentations in addition to the systematical analysis of existing augmentation methods. Due to the above reason, I tend to keep the current score.

---

### Official Review · Reviewer_T7Qz · 2022-10-24

**Confidence:** 3
**Correctness:** 2
**Technical Novelty And Significance:** 2
**Empirical Novelty And Significance:** 2
**Recommendation:** 3

**Clarity, Quality, Novelty And Reproducibility:**

The paper has a fair novelty in its method. However, the paper has a pool of clarity that prevents the reader from fully understanding the proposed method and appreciating this work.

**Strength And Weaknesses:**

Pros:

1. The paper study a novel problem of how data augmentation affects adversarial attacks' effectiveness.

2. The work considers multiple model architectures to test its proposed method.

Cons:

1. Lack of clarity
- 1(a) Section 3.3. The author says “In parallel composition, augmentation methods are applied independently on the input, and their outputs are aggregated (i.e., taking their union)”. I cannot understand, what means by "their outputs are aggregated (i.e., taking their union)". Do you feed the model with the union of augmented images? Or randomly choose one augmentation to apply?
- 1(b) Section 5.1. The section title says "COLOR-SPACE AUGMENTATIONS OUTPERFORM THE STATE OF THE A RT". This title does not match the argument in this paragraph. What this paragraph says is actually, composing COLOR-SPACE AUGMENTATIONS with the SOTA method (DST) is better than the SOTA (DST-MI-FGSM)
- 1(c) Section 5.2. The author says "The results reflected a mostly monotonic relationship between transferability and augmentations." But the results seem to be not displayed in this paper.
- 1(d) Section 5.4 The author says "As can be seen from Table 5, the methods least conducive for transferability (NeuTrans and Sharp) ..." I do not find any evidence saying that NeuTrans and Sharp have the least transferability.


**Summary Of The Paper:**

The paper proposes to enhance the transferability of adversarial examples by doing adversarial attacks using multiple data augmentation parallelly.

**Summary Of The Review:**

Based on the pool clarity, I am afraid that the current submission does not reach the bar of acceptance. If the clarity is improved in the rebuttal, I will consider re-evaluating this manuscript.

---

> ### Author Response · Authors · 2022-11-19
> **Addressing comments and revision changes**
>
> We thank the reviewer for pointing out the lack of clarity which helped us improve the paper's readability. We revised the text as follows to better convey methods and takeaways.
>
> > 1(a) Section 3.3. The author says “In parallel composition, augmentation methods are applied independently on the input, and their outputs are aggregated (i.e., taking their union)”. I cannot understand, what means by "their outputs are aggregated (i.e., taking their union)". Do you feed the model with the union of augmented images? Or randomly choose one augmentation to apply?
>
> We clarified the composition procedures by re-writing the text in Section 3.3 and adding an illustrative figure (Figure 1) to Appendix A. When serially composing augmentations, each augmentation method operates on the output of the previous one. By contrast, in parallel composition, each augmentation method operates independently on the input (or set of inputs), and their outputs are aggregated by taking their union to augment attacks. Thus, in serial composition, the number of samples grows *exponentially* in the number of augmentation methods. As we consider a large number of augmentation methods, using serial composition will lead to prohibitive memory and time requirements. In contrast, in parallel composition, the number of samples only grows *linearly*, making our attack feasible (as is shown in the AE generation time measurements we added in Appendix I). In this work, for consistency with prior work (e.g., [Wang et al. 2021]), we use serial composition when composing diverse inputs (DI), scaling (Sc.), and translations (Tr.). Other augmentation methods are composed in parallel.
>
> > Section 5.1. The section title says "COLOR-SPACE AUGMENTATIONS OUTPERFORM THE STATE OF THE ART". This title does not match the argument in this paragraph. What this paragraph says is actually, composing COLOR-SPACE AUGMENTATIONS with the SOTA method (DST) is better than the SOTA (DST-MI-FGSM)
>
> Thank you for pointing out the imprecise wording. We renamed Section 5.1’s title to “Color-space augmentations significantly advance the state of the art.”
>
> > Section 5.2. The author says "The results reflected a mostly monotonic relationship between transferability and augmentations." But the results seem to be not displayed in this paper.
>
> The newly added Figure 2 (Appendix H) visualizes this monotonic relationship. For your convenience, here’s a link to an interactive version of the figure (best viewed after zooming in):
> https://i.postimg.cc/gdCg7KSP/monoicity.gif. Each node in the figure represents a composition of augmentation methods. The binary string within a node encodes the composition: Each bit, from the most to the least significant, denotes whether Admix (most significant bit), GS, CutOut, AutoAugment, DST-MI-FGSM, NeuTrans, and Sharp (least significant bit), respectively, is included (1) or excluded (0) from the composition. An edge from node u to v is included if v includes exactly one more augmentation method compared to $u$. An edge $(u, v)$ is colored in black (resp. red) if $v$’s composition achieves higher or equal (resp. lower) average transferability rates than $u$ when integrated into MI-FGSM. We fade away nodes containing NeuTrans or Sharp and their corresponding edges. Notice how all the remaining (unfaded) edges are black, showing that the relationship between the (average) transferability and the remaining augmentations is monotonic (i.e., more augmentations composed $\rightarrow$ $\ge$transferability).
>
> > 1(d) Section 5.4 The author says "As can be seen from Table 5, the methods least conducive for transferability (NeuTrans and Sharp) ..." I do not find any evidence saying that NeuTrans and Sharp have the least transferability.
>
> We revised the text (Section 5.4), clarifying that the only augmentation methods that *may fail to improve transferability* harm the DNN benign accuracy.
>
> ## Refs.
>
> [Wang et al., 2021] Wang et al. “Admix: Enhancing the transferability of adversarial attacks.” In Proc. ICCV, 2021.

---

### Official Review · Reviewer_KVr6 · 2022-10-28

**Confidence:** 4
**Correctness:** 4
**Technical Novelty And Significance:** 2
**Empirical Novelty And Significance:** 2
**Recommendation:** 3

**Clarity, Quality, Novelty And Reproducibility:**

The paper overall is well-written and presents most of the matter in a clear, concise manner. While the key contributions are an extension of prior works, the empirical evaluations presented are fairly thorough.

**Strength And Weaknesses:**

Strengths:
1) The paper clearly motivates the study of augmentation to boost adversarial transferability, and covers contributions of past works in a clear, concise manner which helps place the findings uncovered in a systematic framework.
2) The paper presents a fairly detailed set of empirical evaluations on a wide range of deep networks with different architectures, on a 430-class subset of ImageNet with 1000 images that is commonly used to assess adversarial transferability. The inclusion of both normally trained and adversarially trained models in the evaluations helps establish the results in a detailed manner.
3) The proposed combination of augmentations is largely seen to outperform the current state of the art, which largely serves as a baseline, especially considering that DST-MI-FGSM is a subset of the final augmentations utilized.



Weaknesses:
1) With the large number of possible augmentations available, the paper primarily focuses on the parallel composition of augmentations, as opposed to the serial case as done in prior works (eg. as in DST-MI-FGSM). However, given that this is a key aspect of the paper, more details could have been provided in Section-3.3 to better understand the associated pros and cons.
2) For example, it appears that parallel augmentations apriori require additional forward and backward passes as opposed to serial composition. Could the authors kindly clarify if the parameter of interest “m” in Algorithm1 is kept constant for the proposed approach as compared to prior works? Or is the sampling over the distribution D done independently for different augmentations in the parallel-composition setting?
3) Furthermore, given that the prior methods such as DST-MI-FGSM are subsumed by the proposed approach, a more thorough complexity analysis needs to be included. Thus, the running time required for generating the attacks using UltimateCombo could be reported for a subset of the models considered.
4) While the empirical results presented are indeed interesting, the paper could have possibly devoted more attention to analyzing why such compositions are more effective. For example, a slightly more detailed discussion as currently provided in Section-5.4 could have been included with respect to cosine similarity of gradients post parallel-composition, and the direction towards mitigating the effect of overfitting during the use of a single surrogate model as compared to an ensemble.
5) While the majority of the results are presented on the ImageNet subset, additional evaluations over other standard datasets such as CIFAR-10 or CIFAR-100 could be included, given that it has served as a benchmark for adversarial defense research for the past few years (though some prior works on adversarial transferability exclude these as well). This could also potentially help weed out results that are overly-specific to the case of ImageNet based images, and help generate a better understanding towards the compositional nature of augmentations.


Minor Typos-
 Page3: “ set of these defense.” -> ” set of these defenses.”
Page5: “The main difference from e is that” -> “The main difference from RE is that”




**Summary Of The Paper:**

In this work, the authors study in detail the effect of a diverse set of data-augmentation techniques on the transferability of adversarial attacks in the black-box setting. The paper presents a large systematized empirical study of the effect of adversarial transferability with the use of augmentations such as Color Jitter, Fancy Principle Component Analysis, Channel shuffle, Greyscale, random erasing, sharpening, CutMix, neural transfer and AutoAugment. In particular, the paper finds that the composition of several such augmentations generally yields increased transferability.

**Summary Of The Review:**

The paper presents a systematic study regarding the composition of augmentations to boost adversarial transferability. Indeed, it finds a performative subset such that enhanced transferability is observed over prior state-of-the-art approaches, which are largely subsumed by the proposed composition. However, as mentioned in the weaknesses section, more details could be provided to clarify key aspects of the proposed approach, alongside complexity comparisons to prior approaches. I would be willing to raise my score further if these points could be addressed in detail.


Post-Rebuttal Update:
A note has been added titled "Post-Rebuttal Comments and Modification to Recommendation" in the thread below. I would like to update my score to be “3: reject, not good enough".

---

> ### Author Response · Authors · 2022-11-19
> **Addressing comments and revision changes (continued)**
>
> > While the majority of the results are presented on the ImageNet subset, additional evaluations over other standard datasets such as CIFAR-10 or CIFAR-100 could be included, given that it has served as a benchmark for adversarial defense research for the past few years (though some prior works on adversarial transferability exclude these as well). This could also potentially help weed out results that are overly-specific to the case of ImageNet based images, and help generate a better understanding towards the compositional nature of augmentations.
>
> To evidence the generality of our findings, in the revised version, we added new experiments comparing our method to previous work across different datasets, perturbation strengths, and defenses. All experiments support our claim and show significant improvement in transferability compared to previous state-of-the-art methods. The new settings we tested are:
>
> * CIFAR-10 using different architectures (Tables 14 and 15).
> * Different epsilon values with ImageNet (Table 13)
> * Different epsilon values with CIFAR-10 (Tables 14 and 15)
> * Evaluating TRS, a state-of-the-art defense tailored for transferability-based attacks [Yang et al., 2021] (Table 5).
>
> ## Refs.
>
> [Lin et al., 2020] Lin et al. “Nesterov accelerated gradient and scale invariance for adversarial attacks.” In Proc. ICLR, 2020.
>
> [Wang et al., 2021] Wang et al. “Admix: Enhancing the transferability of adversarial attacks.” In Proc. ICCV, 2021.
>
> [Yang et al., 2021] Yang et al. “TRS: Transferability reduced ensemble via encouraging gradient diversity and model smoothness.” In Proc. NeurIPS, 2021.

---

> ### Author Response · Authors · 2022-11-19
> **Addressing comments and revision changes**
>
> We thank the reviewer for the detailed and constructive comments. We addressed the reviewer feedback as described below.
>
> > With the large number of possible augmentations available, the paper primarily focuses on the parallel composition of augmentations, as opposed to the serial case as done in prior works (eg. as in DST-MI-FGSM). However, given that this is a key aspect of the paper, more details could have been provided in Section-3.3 to better understand the associated pros and cons.
>
> We clarified the composition procedures by re-writing the text in Section 3.3 and adding an illustrative figure (Figure 1) to Appendix A. When serially composing augmentations, each augmentation method operates on the output of the previous one. By contrast, in parallel composition, each augmentation method operates independently on the input (or set of inputs), and their outputs are aggregated by taking their union to augment attacks. Thus, in serial composition, the number of samples grows *exponentially* in the number of augmentation methods. As we consider a large number of augmentation methods, using serial composition will lead to prohibitive memory and time requirements. In contrast, in parallel composition, the number of samples only grows *linearly*, making our attack feasible (as is shown in the AE generation time measurements we added in Appendix I). In this work, for consistency with prior work (e.g., [Wang et al. 2021]), we use serial composition when composing diverse inputs (DI), scaling (Sc.), and translations (Tr.). Other augmentation methods are composed in parallel.
>
> > For example, it appears that parallel augmentations apriori require additional forward and backward passes as opposed to serial composition. Could the authors kindly clarify if the parameter of interest “m” in Algorithm1 is kept constant for the proposed approach as compared to prior works? Or is the sampling over the distribution D done independently for different augmentations in the parallel-composition setting?
>
> Different compositions (and composition types) produce a different number of samples to augment attacks. Hence, consistently with prior work (e.g., Lin et al. (2020) and Wang et al. (2021)), $m$ in Algorithm 1 varies, depending on the composition of augmentations. This is also reflected in attack run-times (Appendix I), where augmentations techniques and compositions that produce more samples exhibit slower attack run-time.
>
> > Furthermore, given that the prior methods such as DST-MI-FGSM are subsumed by the proposed approach, a more thorough complexity analysis needs to be included. Thus, the running time required for generating the attacks using UltimateCombo could be reported for a subset of the models considered.
>
> Appendix I (chiefly, Table 17) in the revision analyzes the running time of our attack. We also added summary results for running time to Sections 5.1 and 5.3. In a nutshell, we found that GS-DST-MI-FGSM is slower than DST-MI-FGSM (as expected) but faster than all other attacks. UltimateCombo was slower than DST-MI-FGSM and Admix-DT-MI-FGSN, but substantially faster than DST-VMI-FGSM.
>
> > While the empirical results presented are indeed interesting, the paper could have possibly devoted more attention to analyzing why such compositions are more effective. For example, a slightly more detailed discussion as currently provided in Section-5.4 could have been included with respect to cosine similarity of gradients post parallel-composition, and the direction towards mitigating the effect of overfitting during the use of a single surrogate model as compared to an ensemble.
>
> We are pleased to find out that the reviewer thought the results were interesting. We are uncertain that we understand the reviewer’s suggestion for extending the analysis. If the reviewer could kindly clarify, we would gladly incorporate his suggestion.

---

### Official Review · Reviewer_DKMA · 2022-10-31

**Confidence:** 4
**Correctness:** 3
**Technical Novelty And Significance:** 2
**Empirical Novelty And Significance:** 2
**Recommendation:** 3

**Clarity, Quality, Novelty And Reproducibility:**

The paper wastes too much contents on backgrounds, which hurts the presentation clarity, and the novelty is very limited. The reproducibility is good as the codes are provided via an anonymous link.

**Strength And Weaknesses:**

**Strength**

1. An empirical investigation about the combination of the best practices in data augmentations for boosting adversarial transferability can provide useful insights for the community.

**Weakness**

1. The novelty of this paper is very limited. It manually combines existing data augmentations to generate a better augmentation strategy without providing theoretical analysis or an automatic solution, which is more like a technical report. The technical contributions cannot match the bar of ICLR.

2. The paper wastes too much contents on background knowledge about adversarial attacks and data augmentations, and starts to introduce the experimental results, which are the major contributions of this paper, from the 6th page. The presentation style can be better organized.

3. The experimental results are not solid enough. Only one dataset with 1000 images is considered, while the datasets like CIFAR-10/100/SVHN/ImageNet, which are commonly adopted in the literature, are not included. In addition, only one perturbation strength (16/255) is considered. It is highly desired to conduct experiments across different datasets, perturbation strengths, and defensive methods, otherwise the accuracy gap between the proposed methods and baselines in Table 1/2 may be overturned via tuning the hyperparameters of adversarial example generation.

**Summary Of The Paper:**

Driven by the success of data augmentation in improving the transferability of adversarial samples, this paper conducts an empirical investigation about the best combination of different data augmentations towards boosting the adversarial transferability. Experiments across ten data augmentations lead to a new composition that outperforms previous data augmentation methods.

**Summary Of The Review:**

Considering the limited novelty and the lack of necessary experiments as elaborated in the weakness section, I tend to reject this paper.

---

> ### Author Response · Authors · 2022-11-19
> **Addressing comments and changes**
>
> We thank the reviewer for the helpful feedback.
>
> > The novelty of this paper is very limited. It manually combines existing data augmentations to generate a better augmentation strategy without providing theoretical analysis or an automatic solution, which is more like a technical report. The technical contributions cannot match the bar of ICLR.
>
> Our work is the first to systematically analyze and compare the contribution of a wide range of data-augmentation techniques to transferability. Although our work relies on existing schemes originally proposed to help ML models generalize, we show that by composing these, we can significantly improve the start-of-the-art in transferability. We agree that introducing new theories that can help explain our results or developing data augmentation schemes optimized for transferability are important open problems that result from our work. However, we believe that highlighting the usefulness of our approach and the importance of these questions is a significant first step. In the revised version (Section 6), we added a description of the open questions. We hope our work will motivate further research in these directions.
>
> >The paper wastes too much contents on background knowledge about adversarial attacks and data augmentations, and starts to introduce the experimental results, which are the major contributions of this paper, from the 6th page. The presentation style can be better organized.
>
> As suggested, we rewrote the background to make it more concise. We note that we still tried to keep the paper self-contained and also clarify questions raised by the reviewers.
>
>
> > The experimental results are not solid enough. Only one dataset with 1000 images is considered, while the datasets like CIFAR-10/100/SVHN/ImageNet, which are commonly adopted in the literature, are not included. In addition, only one perturbation strength (16/255) is considered. It is highly desired to conduct experiments across different datasets, perturbation strengths, and defensive methods, otherwise the accuracy gap between the proposed methods and baselines in Table 1/2 may be overturned via tuning the hyperparameters of adversarial example generation.
>
> To evidence the generality of our findings, in the revised version, we added new experiments comparing our method to previous work across different datasets, perturbation strengths, and defenses. All experiments support our claim and show significant improvement in transferability compared to previous state-of-the-art methods. The new settings we tested are:
>
> * CIFAR-10 using different architectures (Tables 14 and 15).
> * Different epsilon values with ImageNet (Table 13)
> * Different epsilon values with CIFAR-10 (Tables 14 and 15)
> * Evaluating TRS, a state-of-the-art defense tailored for transferability-based attacks [Yang et al., 2021] (Table 5).
>
> ## Refs.
>
> [Yang et al., 2021] Yang et al. “TRS: Transferability reduced ensemble via encouraging gradient diversity and model smoothness.” In Proc. NeurIPS, 2021.

---

### Author Response · Authors · 2022-11-19
**Thank you and revision summary**

We thank the reviewers for their constructive feedback! We worked extensively to address the reviewers’ comments and suggestions. In particular, we have made the following additions and changes in the revision:

1. Introduced comparisons with an additional baseline (DST-VMI-FGSM in Sections 4 and 5)
2. Added experiments with the CIFAR-10 dataset and additional neural network architectures (Tables 14 and 15)
3. Included experiments with different epsilon values for CIFAR-10 (Tables 14 and 15) and ImageNet (Table 13)
4. Evaluated attacks against TRS, a state-of-the-art defense tailored for transferability-based attacks (Table 5)
5. Added a figure to visualize the monotonic relationship between augmentation methods and transferability (Figure 2 in Appendix H)
6. Included run-time measurements for attacks (Appendix I, with summaries in Sections 5.1 and 5.3)
7. Clarified how parallel and serial compositions operate (Section 3.3 and Appendix A)
8. Discussed other transferability-based attacks excluded from the evaluation (Appendix D)
9. Raised open questions for future work (Section 6)
10. Rewrote the background to make it more concise and make space for additional results
11. Made additional minor writing tweaks to make claims more precise

Overall, we found that the new experimental results further the validity of the paper’s claims, and believe that the revision is more readable and easier to understand. Below, we address each reviewer comment in detail. We hope that the revision addresses the reviewers’ concerns.

---

### Decision · Program_Chairs · 2023-01-20

**Decision:**

Reject

**Justification For Why Not Higher Score:**

limited novelty

**Justification For Why Not Lower Score:**

n/a

**Metareview: Summary, Strengths And Weaknesses:**

This paper experimentally verifies the effect of augmentation (and their compositions) in improving the transferability of adversarial examples.
However, all of the reviewers insisted on rejecting this paper due to several reasons. The biggest concern is that the novelty of this paper is very limited. The fact that performance can be improved simply through known augmentations is a fact that can be generally accepted in almost all tasks. I would like to recommend that you analyze the nature of this task in more depth and propose augmentation specific to this task.